# CompoDiff: Versatile Composed Image Retrieval With Latent Diffusion

**Geonmo Gu**[*, 1]**, Sanghyuk Chun**[*, 2]**, Wonjae Kim**[2]**, HeeJae Jun**[1]**, Yoohoon Kang**[1]**, Sangdoo Yun**[2]

[1]NAVER Vision      [2]NAVER AI Lab      [*] Equal contribution

**Reviewed on OpenReview:** https://openreview.net/forum?id=mKtlzWObWc

## Abstract

This paper proposes a novel diffusion-based model, CompoDiff, for solving zero-shot Composed Image Retrieval (ZS-CIR) with latent diffusion. This paper also introduces a new synthetic dataset, named SynthTriplets18M, with 18.8 million reference images, conditions, and corresponding target image triplets to train CIR models. CompoDiff and SynthTriplets18M tackle the shortages of the previous CIR approaches, such as poor generalizability due to the small dataset scale and the limited types of conditions. CompoDiff not only achieves a new state-of-the-art on four ZS-CIR benchmarks, including FashionIQ, CIRR, CIRCO, and GeneCIS, but also enables a more versatile and controllable CIR by accepting various conditions, such as negative text, and image mask conditions. CompoDiff also shows the controllability of the condition strength between text and image queries and the trade-off between inference speed and performance, which are unavailable with existing CIR methods. The code and dataset are available at https://github.com/navervision/CompoDiff.

## 1 Introduction

Imagine a customer seeking a captivating cloth serendipitously found on social media but not the most appealing materials and colors. In this scenario, the customer needs a search engine that can process composed queries, *e.g.*, the reference garment image along with text specifying the preferred material and color. This task has been recently formulated as *Composed Image Retrieval (CIR)*. CIR systems offer the benefits of searching for visually similar items while providing a high degree of freedom to depict text queries as text-to-image retrieval. CIR can also improve the search quality by iteratively taking user feedback.

The existing CIR methods address the problem by combining image and text features using additional fusion models, *e.g.*, $z_i = \texttt{fusion}(z_{i_R}, z_c)$ where $z_i$, $z_c$, $z_{i_R}$ are the target image, conditioning text, and reference image features, respectively[1]. Although the fusion methods have shown great success, they have fundamental limitations. First, the fusion module is not flexible; it cannot handle versatile conditions beyond a limited textual one. For instance, a user might want to include a negative text that is not desired for the search ($x_{c_{T^-}}$) (*e.g.*, an image + "with cherry blossom" − "France", as in Fig. 1 (b)), indicate where ($x_{c_M}$) the condition is applied (*e.g.*, an image + "balloon" + indicator, as in Fig. 1 (c)), or construct a complex condition with a mixture of them. Furthermore, once the fusion model is trained, it will always produce the same $z_i$ for the given $z_{i_R}$ and $z_c$ to users. However, a practical retrieval system needs to control the strength of conditions by its applications or control the level of serendipity. Second, they need a pre-collected human-verified dataset of triplets $\langle x_{i_R}, x_c, x_i \rangle$ consisting of a reference image ($x_{i_R}$), a text condition ($x_c$), and the corresponding target image ($x_i$). However, obtaining such triplets is costly and sometimes impossible; therefore, the existing CIR datasets are small-scale (*e.g.*, 30K triplets for Fashion-IQ (Wu et al., 2021) and 36K triplets for CIRR (Liu et al., 2021)), resulting in a lack of generalizability to other datasets.

---

[1]Throughout this paper, we will use $x$ to denote raw data and $z$ to denote a vector encoded from $x$.

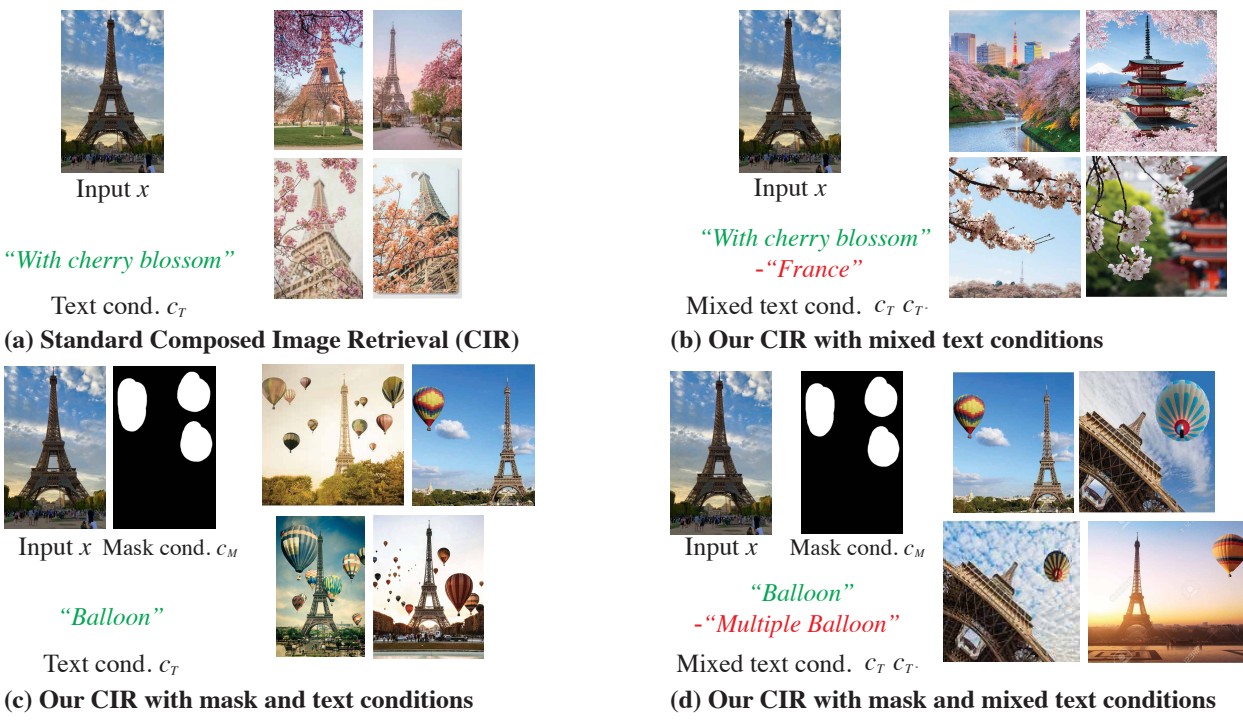

Figure 1: **Composed Image Retrieval (CIR) scenarios.** (a) A standard CIR scenario. (b-d) Our versatile CIR scenarios with mixed conditions (*e.g.*, negative text and mask). Results by CompoDiff on LAION-2B.

We aim to achieve a generalizable CIR model with diverse and versatile conditions by using latent diffusion. We treat the CIR task as a conditional image editing task on the latent space, *i.e.*, $z_i = \texttt{Edit}(z_{i_R}|z_c, \dots)$. Our diffusion-based CIR model, named CompoDiff, can easily deal with versatile and complex conditions, benefiting from the flexibility of the latent diffusion model (Rombach et al., 2022) and the classifier-free guidance (Ho & Salimans, 2022). Note that the purpose of image editing tasks and CIR tasks are different; an image editing model aims to find any image satisfying the changes due to the instruction, while a CIR model aims to find a generalized concept of the given composed queries. For example, assume a picture of a dog on a grass field and an instruction "change the dog to a cat". Here, even if we assume a perfect image generator can generate an image of a cat on a grass field, it could not achieve a good retrieval performance because it cannot generate all possible cat images in the database. In other words, the purpose of image editing is "precision", while the purpose of image retrieval is "recall"; it makes the difference between image editing and retrieval. In our experiments, we support this claim by showing directly using an editing model performs much worse than other CIR methods. Therefore, we need a specialized model for CIR rather than directly using the image editing model. In this paper, we tackle the problem by using a latent diffusion model instead of an image-level diffusion model and introducing various retrieval-specific conditions, such as mask conditions. We train a latent diffusion model that translates the embedding of the reference image ($z_{i_R}$) into the embedding of the target image ($z_i$) guided by the embedding of the given text condition ($z_c$). As shown in Fig. 1, CompoDiff can handle various conditions, which is not possible with the standard CIR scenario with the limited text condition $x_{c_T}$. Although our method has an advantage over the existing fusion-based CIR methods in terms of versatility, CompoDiff also needs to be trained with triplet datasets where the scale of the existing CIR datasets is extremely small.

We address the dataset scale issue by synthesizing a vast set of high-quality **18.8M** triplets of $\langle x_{i_R}, x_c, x_i \rangle$. Our approach is fully automated without human verification; hence, it is scalable even to 18.8M. We follow InstructPix2Pix (IP2P) (Brooks et al., 2022) for synthesizing triplets, while our dataset contains ×40 more triplets and ×12.5 more keywords (*e.g.*, objects, background details, or textures) than IP2P. Our massive dataset, named **SynthTriplets18M**, is over 500 times larger than existing CIR datasets and covers a diverse and extensive range of conditioning cases, resulting in a notable performance improvement for any

CIR model. For example, ARTEMIS (Delmas et al., 2022) trained exclusively with SynthTriplets18M shows outperforming zero-shot performance even than its FashionIQ-trained counterpart (40.6 vs. 38.2).

To show the generalizability of the models, we evaluate the models on the "zero-shot" (ZS) CIR scenario using four CIR benchmarks: FashionIQ (Wu et al., 2021), CIRR (Liu et al., 2021), CIRCO (Baldrati et al., 2023), and GeneCIS (Vaze et al., 2023); *i.e.*, we report the retrieval results by the models trained on our SynthTriplets18M and a large-scale image-text paired dataset without access to the target triplet datasets. In all experiments, CompoDiff achieves the best zero-shot performances with significant gaps (See Table 3). Moreover, we observe that the fusion-based approaches solely trained on SynthTriplets18M (*e.g.*, Combiner (Baldrati et al., 2022)) show comparable or outperforming zero-shot CIR performances compared to the previous SOTA zero-shot CIR methods, *e.g.*, Pic2Word (Saito et al., 2023), SEARLE (Baldrati et al., 2023) and LinCIR (Gu et al., 2024). Furthermore, we qualitatively observe that the retrieval results of CompoDiff are semantically better than previous zero-shot CIR methods, such as Pic2Word, on a large-scale image database, *e.g.*, LAION-2B (Schuhmann et al., 2022a).

Another notable advantage of CompoDiff is the controllability of various conditions during inference, which is inherited from the nature of diffusion models. Users can adjust the weight of conditions to make the model focus on their preference. Users can also manipulate randomness to vary the degree of serendipity. In addition, CompoDiff can control the speed of inference with minimal sacrifice in retrieval performance, accomplished by adjusting the number of sampling steps in the diffusion model. As a result, CompoDiff can be deployed in various scenarios with different computational budgets. All of these controllability features are achievable by controlling the inference parameters of classifier-free guidance without any model training.

Our contributions are first to show the effectiveness of diffusion models on non-generative tasks, such as multi-modal retrieval and data generation. More specifically: (1) We propose a diffusion-based CIR method, named CompoDiff. CompoDiff can handle various conditions, such as the negative text condition, while the previous methods cannot. We also can control the strength of conditions, *e.g.*, more focusing on the reference image while the changes by the text should be small. (2) We generate SynthTriplets18M, a diverse and massive synthetic dataset of 18M triplets that can make CIR models achieve zero-shot (ZS) generalizability. Our data generation process is fully automated and easily scalable. (3) Our experiments support the effectiveness of CompoDiff quantitatively (significant ZS-CIR performances on FashionIQ, CIRR, CIRCO, and GeneCIS datasets) and qualitatively (ZS-CIR on a billion-scale DB, or image decoding using unCLIP generator).

## 2 Related Works

**Composed image retrieval.** The mainstream CIR models have focused on *multi-modal fusion methods*, which combine image and text features extracted from separate visual and text encoders, as in Fig. 2 (a). For example, Vo et al. (2019) and Yu et al. (2020) used CNN and RNN, and Chen et al. (2022) and Anwaar et al. (2021) used CNN and Transformer. More recent methods address CIR tasks by leveraging external knowledge from models pre-trained on large-scale datasets such as CLIP (Radford et al., 2021). For example, Combiner (Baldrati et al., 2022) fine-tunes the CLIP text encoder on the triplet dataset to satisfy the relationship of $z_i = z_{i_R} + z_c$, and then trains a Combiner module on top of the encoders. However, these fusion methods still require expensive pre-collected and human-verified triplets of the target domain.

To solve this problem, recent studies aim to solve CIR tasks in a zero-shot manner. *Inversion-based zero-shot CIR methods*, such as Pic2Word (Saito et al., 2023), SEARLE (Baldrati et al., 2023) and LinCIR (Gu et al., 2024), tackle the problem through text-to-image retrieval tasks where the input image is projected into the condition text – Fig. 2 (b). Note that our zero-shot CIR scenario is slightly different from theirs; our zero-shot CIR denotes that the models are not trained on the target triplet datasets, but trained on our synthetic dataset, SynthTriplets18M, and image-text paired dataset, *e.g.*, LAION-2B (Schuhmann et al., 2022a). Meanwhile, Saito et al. (2023), Baldrati et al. (2023) and Gu et al. (2024) use the term zero-shot when the CIR models are trained without a triplet dataset.

All the existing CIR models only focus on text conditions ($z_{c_T}$) (*e.g.*, Fig. 1 (a)) and have difficulties in handling versatile scenarios (*e.g.*, Fig. 1 (b-d)) with a lack of the controllability. On the other hand, our method enables multiple various conditions and controllabilities with strong zero-shot performances by

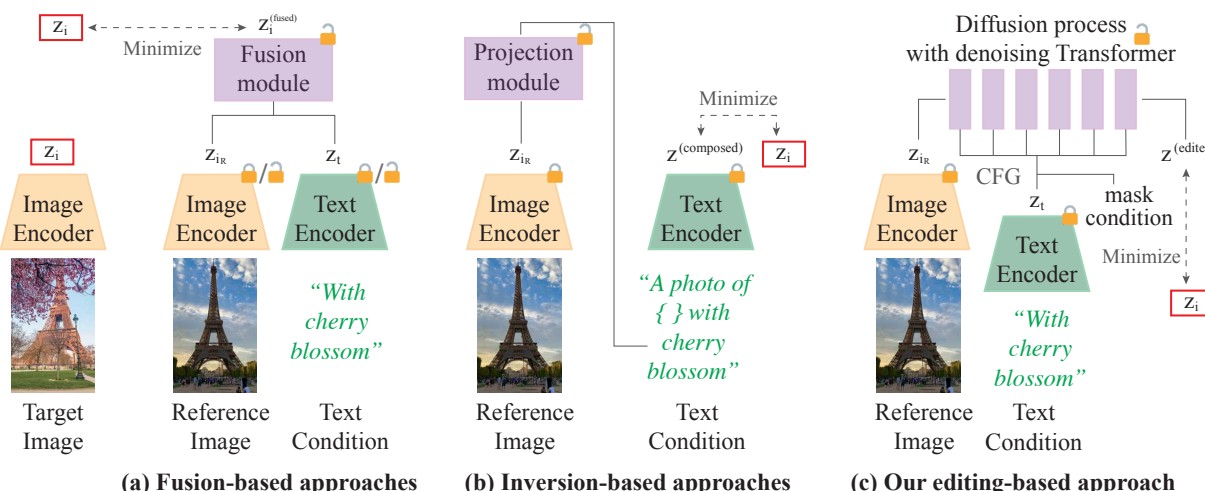

Figure 2: **Comparisons of CIR methods.** (a) Fusion-based methods (*e.g.*, ARTEMIS (Delmas et al., 2022) and Combiner (Baldrati et al., 2022)) make a fused feature from image feature $z_{i_R}$ and text feature $z_c$. (b) Inversion-based methods (*e.g.*, Pic2Word (Saito et al., 2023), SEARLE (Baldrati et al., 2023) and LinCIR (Gu et al., 2024)) project $z_{i_R}$ into the text space, then perform text-to-image retrieval. (c) We apply a diffusion process to $z_{i_R}$ with classifier-free guidance by additional conditions. (b) and (c) use frozen encoders, and (a) usually tunes the encoders.

employing (1) a latent diffusion model (Rombach et al., 2022) with classifier-free guidance (Ho & Salimans, 2022) and (2) a massive high-quality synthetic dataset, SynthTriplets18M.

Concurrent with our work, CoVR (Ventura et al., 2024), VDG (Jang et al., 2024), and MagicLens (Zhang et al., 2024) consider constructing training triplets from the existing image datasets, namely, collecting image pairs and synthesizing the modification texts using LLMs. Although they showed empirically good performances, their approaches need real images where it is often very difficult to generate plausible modification instructions between two real images. Meanwhile, our synthetic dataset is more controllable than their datasets. Namely, we can exactly guide the visual difference between two images with the given instructions.

**Dataset creation with diffusion models.** A conventional data collection process for $\langle x_{i_R}, x_c, x_i \rangle$ is two-staged: collecting candidate reference-target image pairs and manually annotating the modification sentences by human annotators. For example, FashionIQ (Wu et al., 2021) collects the candidate pairs from the same item category (*e.g.*, shirt, dress, and top) and manually annotates the relative captions by crowd workers. CIRR (Liu et al., 2021) gathers the candidate pairs from real-life images from NLVR$^2$ (Suhr et al., 2018). The triplet collection process inevitably becomes expensive, making it difficult to scale CIR datasets. We mitigate this problem by generating a massive synthetic dataset instead of relying on human annotators.

Recently, there have been attempts to generate synthetic data to improve model performance (Brooks et al., 2022; Nair et al., 2022; Shipard et al., 2023) by exploiting the powerful generation capabilities of diffusion models. In particular, Brooks et al. (2022) proposes a generation process for $\langle x_{i_R}, x_c, x_i \rangle$ to train an image editing model. We scale up the dataset synthesis process of Brooks et al. (2022) from 1M triplets to 18.8M. Our generation process contains more diverse triplets by applying the object-level modification process, resulting in better CIR performances on the same 1M scale and easily scalable to a larger number of samples.

# 3 CompoDiff: Composed Image Retrieval with Latent Diffusion

This section introduces Composed Image Retrieval with Latent Diffusion (CompoDiff) which employs a diffusion process in the frozen CLIP latent feature space with classifier-free guidance. Unlike previous latent diffusion models, we use a Transformer-based denoiser. We train our model with various tasks, such as text-to-image (T2I) generation, masked T2I and triplet-based generation to handle various conditions (*e.g.*, negative text or mask), while the previous CIR methods only limit beyond the positive text instruction.

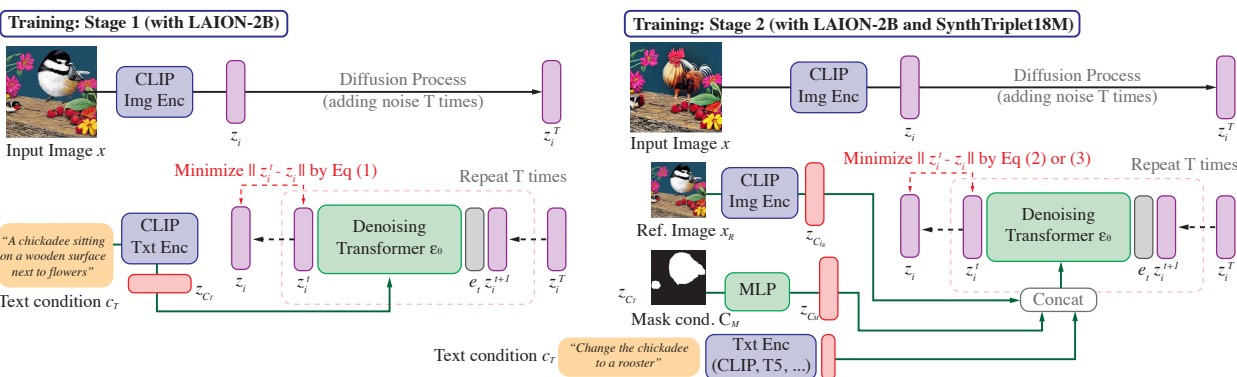

Figure 3: **Training overview.** Stage 1 is trained on LAION-2B with Eq. (1). For stage 2, we alternatively update Denoising Transformer $\epsilon_\theta$ on LAION-2B with Eq. (1) and 2 and SynthTriplets18M with Eq. (3).

## 3.1 Preliminary: Diffusion model

Given a data sample from the distribution $(x \sim p(x))$ diffusion model (DM) defines a Markov chain of latent variables $z_1, \ldots, z_T$ as: $q(z_t|z_{t-1}) = \mathcal{N}(z_t; \sqrt{\alpha_t} z_{t-1}, (1-\alpha_t)I)$. It is proven that (1) the posterior $q_{z_{t-1}|z_t}$ is approximated to a Gaussian distribution with a diagonal covariance, and (2) if the size of chain $T$ is sufficiently large, then $z_T$ will be approximated to $\mathcal{N}(0, I)$. Namely, DMs are a general probabilistic method that maps an input distribution $p(x)$ to a normal distribution $\mathcal{N}(0, I)$, without any assumption on the input data $x$. From this observation, Rombach et al. (2022) proposed to learn DMs on a latent space (*i.e.*, latent diffusion), which brings a huge improvement in efficiency. In practice, we train a denoising module $\epsilon_\theta(z_t, t)$ (*i.e.*, $z_{t-1} = \epsilon_\theta(z_t, t)$) where the timestamp $t$ is conditioned by employing time embeddings $e_t$.

The recent DMs use classifier-free guidance (CFG) (Ho & Salimans, 2022) for training conditional DMs without a pre-trained classifier (Dhariwal & Nichol, 2021). In this paper, we provide various conditions via CFG to take two advantages. First, we can easily control the intensity of the condition. Second, we can extend the conditions beyond a single text condition, *e.g.*, a negative text condition or a mask condition.

CompoDiff has two distinct differences from the previous latent diffusion approaches, such as StableDiffusion (SD) (Rombach et al., 2022). First, SD performs the diffusion process on $64 \times 64$-dimensional VQ-GAN (Esser et al., 2021) latent space. Meanwhile, the diffusion process of CompoDiff is performed on the CLIP latent embedding space. Therefore, the edited features by CompoDiff can be directly used for retrieval on the CLIP space, while SD cannot. The Dall-e2 prior (Ramesh et al., 2022) performs the diffusion process on the CLIP embedding space, but CompoDiff takes inputs and conditions differently, as described below.

As the second contribution, CompoDiff uses a different architecture for the de-noising module. While SD and other DM models are based on U-Net structure (Ronneberger et al., 2015), CompoDiff uses a simple Transformer (Vaswani et al., 2017). Moreover, CompoDiff is designed to handle multiple conditions, such as masked conditions, and a triplet relationship. SD cannot handle the localized condition and is designed for a pairwise relationship (*e.g.*, text-to-image generation). CompoDiff also handles the condition different from Dalle-2 prior. Dalle-2 prior handles conditions as the input of the diffusion model, but our CompoDiff diffusion Transformer takes the conditions via the cross-attention mechanism. This design choice makes the inference speed of CompoDiff faster. If the conditions are concatenated to the input tokens, the inference speed will be highly degenerated (Song et al., 2022). Table 8 shows that the structure taking all conditions as the concatenated input (*e.g.*, Dalle-2 prior-like) is three times slower than our cross-attention approach.

## 3.2 Training

CompoDiff uses a two-stage training strategy as illustrated in Fig. 3. In stage 1, we train a text-to-image latent diffusion model on LAION-2B. In stage 2, we fine-tune the model on our synthetic triplet dataset, SynthTriplets18M, and LAION-2B. Below, we describe the details of each stage.

In **stage 1**, we train a transformer decoder to convert CLIP textual embeddings into CLIP visual embeddings. This stage is similar to training the Dalle-2 prior, but our model takes only two tokens; a noised CLIP image embedding and a diffusion timestep embedding. The Dalle-2 prior model is computationally inefficient because it also takes 77 encoded CLIP text embeddings as an input. However, CompoDiff uses the encoded text embeddings as conditions through cross-attention mechanisms, which speeds up the process by a factor of three while maintaining similar performance (See Section 5.7). Instead of using the noise prediction of Ho et al. (2020), we train the transformer decoder to predict the denoised $z_i$ directly due to the stability.

Now, we introduce the objective of the first stage with CLIP image embeddings of an input image $z_i$, encoded CLIP text embeddings for text condition $z_{c_T}$, and the denoising Transformer $\epsilon_\theta$:

$$\mathcal{L}_{\text{stage1}} = \mathbb{E}_{t \sim [1,T]} \|z_i - \epsilon_\theta(z_i^{(t)}, t|z_{c_T})\|^2 \tag{1}$$

During training, we randomly drop the text condition by replacing $z_{c_T}$ with a null text embedding $\varnothing_{c_T}$ in order to induce CFG. We use the empty text CLIP embedding ("") for the null embedding.

In **stage 2**, we incorporate condition embeddings, injected by cross-attention, into CLIP text embeddings, along with CLIP reference image visual embeddings and mask embeddings (See Fig. 3). We fine-tune the model with three different tasks: a conversion task that converts textual embeddings into visual embeddings (Eq. (1)), a mask-based conversion task (Eq. (2)), and the triplet-based CIR task (Eq. (3)). The first two tasks are trained on LAION-2B and the last one is trained on SynthTriplets18M.

**The mask-based conversion task** learns a diffusion process that recovers the full image embedding from a masked image embedding. As we do not have mask annotations, we extract masks using a zero-shot text-conditioned segmentation model, CLIPSeg (Lüddecke & Ecker, 2022). We use the nouns of the given caption for the CLIPSeg conditions. Then, we add a Gaussian random noise to the mask region of the image and extract $z_{i,\text{masked}}$. We also introduce mask embedding $z_{c_M}$ by projecting a 64×64 resized mask to the CLIP embedding dimension using a MLP. Now, the mask-based conversion task is defined as follows:

$$\mathcal{L}_{\text{stage2\_masked\_conversion}} = \mathbb{E}_{t \sim [1,T]} \|z_i - \epsilon_\theta(z_{i,\text{masked}}^{(t)}, t|z_{c_T}, z_{i,\text{masked}}, z_{c_M})\|^2, \tag{2}$$

Finally, we introduce the triplet-based training objective to solve CIR tasks on SynthTriplets18M as follows:

$$\mathcal{L}_{\text{stage2\_triplet}} = \mathbb{E}_{t \sim [1,T]} \|z_{i_T} - \epsilon_\theta(z_{i_T}^{(t)}, t|z_{c_T}, z_{i_R}, z_{c_M})\|^2, \tag{3}$$

where $z_{i_R}$ is a reference image feature and $z_{i_T}$ is a modified target image feature.

We update the model by randomly using one of Eq. (1), Eq. (2), and Eq. (3) with the proportions 30%, 30%, 40%. As stage 1, the stage 2 conditions are randomly dropped except for the mask conditions. We use an all-zero mask condition for the tasks that do not use a mask condition. When we drop the image condition of Eq. (3), we replace $z_{i_R}$ with the null image feature, an all zero vector.

Our two-staged training strategy is not a mandatory procedure, but it brings better performances. Conceptually, stage 1 is a pre-training of a diffusion model (DM) with pairwise image-text relationships. The stage 2 is for a fine-tuning process using triplet relationships. Lastly, the alternative optimization strategy for stage 2 is for helping optimization, not for resolving the instability. As shown in Table 8, CompoDiff can be trained only with Eq. (3), but adding more objective functions makes the final performances stronger. The two-staged training strategy is also widely used for the other CIR methods, such as SEARLE (Baldrati et al., 2023) or Combiner (Baldrati et al., 2022). In terms of diffusion model fine-tuning, we argue that our method is not specifically complex than other fine-tuning methods, such as ControlNet (Zhang et al., 2023).

### 3.3 Inference

Fig. 4 shows the overview. Given a reference image feature $z_{i_R}$, a text condition feature $z_{c_T}$, and a mask embedding $z_{c_M}$, we apply a denoising diffusion process based on CFG (Ho & Salimans, 2022) as follows:

$$\begin{aligned}
\tilde{\epsilon}_\theta(z_i^{(t)}, t|z_{c_T}, z_{i_R}, z_{c_M}) = {} & \epsilon_\theta(z_i^{(t)}, t|\varnothing_{c_T}, \varnothing_{i_R}, z_{c_M}) + w_I(\epsilon_\theta(z_i^{(t)}, t|\varnothing_{c_T}, z_{i_R}, z_{c_M}) - \epsilon_\theta(z_i^{(t)}, t|\varnothing_{c_T}, \varnothing_{i_R}, z_{c_M})) \\
& + w_T(\epsilon_\theta(z_i^{(t)}, t|z_{c_T}, z_{i_R}, z_{c_M}) - \epsilon_\theta(z_i^{(t)}, t|\varnothing_{c_T}, z_{i_R}, z_{c_M}))
\end{aligned} \tag{4}$$

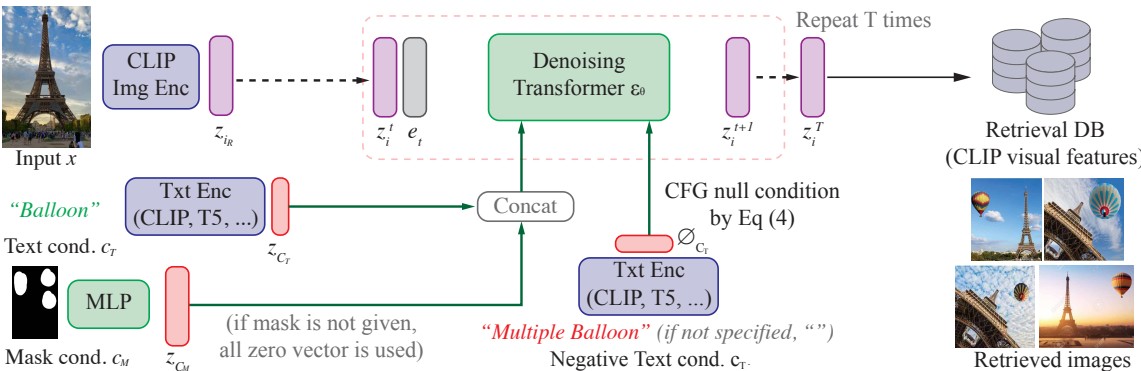

Figure 4: **Inference overview.** Using the denoising transformer $\varepsilon_\theta$ trained by Stage 1 and 2 (Fig. 3), we perform composed image retrieval (CIR). We use the classifier-free guidance by Eq. (4) to transform the input reference image to the target image feature, and perform image-to-image retrieval on the retrieval DB.

where $\varnothing$ denotes null embeddings, *i.e.*, the empty text ("") CLIP textual embedding for the text null embedding and an all-zero vector for the image null embedding. $w_I$ and $w_T$ are weights for controlling the effect of image query or text query, respectively. We can control the degree of the visual similarity ($w_I$) or the text instruction ($w_T$) by simply adjusting the weight values without any training. One of the advantages of Eq. (4) is the ability to handle various conditions at the same time. When using negative text, we simply replace $\varnothing_{i_T}$ with the CLIP text embeddings $c_{T^-}$ for the negative text. These advantages are not available by the previous CIR methods without additional training.

Another advantage of CFG is the controllability of the queries without training, *e.g.*, it allows to control the degree of focus on image features to preserve the visual similarity with the reference by simply adjusting the weights $w_I$ or $w_T$. We show the top-1 retrieved item by varying the image and text weights $w_I$ and $w_T$ from LAION-2B in Fig. 5. By increasing $w_I$, CompoDiff behaves more like an image-to-image retrieval model. Increasing $w_T$, on the other hand, makes CompoDiff focus more on the "pencil sketch" text condition. We use $(w_I, w_T) = (1.5, 7.5)$ for our experiments. The full retrieval performance by varying $w_I$ and $w_T$ is shown in Fig. 10.

As our model is based on a diffusion process, we can easily control the balance between the inference time and the retrieval quality of the modified feature by varying step size. In practice, we set the step size to 5 or 10, which shows the best trade-off.

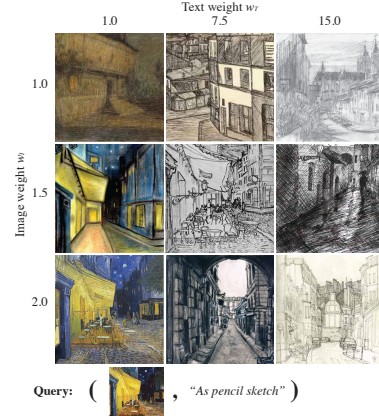

Figure 5: Inference condition control by varying $w_I$, $w_T$ in Eq. (4).

## 4 SynthTriplets18M: Massive High-Quality Synthesized Dataset

CIR requires a dataset of triplets $\langle x_{i_R}, x_c, x_i \rangle$ of a reference image ($x_{i_R}$), a condition ($x_c$), and the corresponding target image ($x_i$). Instead of collecting a dataset by humans, we propose to automatically generate massive triplets by using generative models. We follow the main idea of Instruct Pix2Pix (IP2P) (Brooks et al., 2022). First, we generate $\langle x_{t_R}, x_c, x_t \rangle$ where $x_{t_R}$ is a reference caption, $x_c$ is a modification instruction text, and $x_t$ is the caption modified by $x_c$. We use two strategies to generate $\langle x_{t_R}, x_c, x_t \rangle$: (1) We collect massive captions from the existing caption datasets and generate the mod-

|  | IP2P | SynthTriplets18M |
|---|---|---|
| $\langle x_{t_R}, x_c, x_t \rangle$ (before filtering) | 452k | 500M |
| $\langle x_{t_R}, x_c, x_t \rangle$ (after filtering) | 313k | 60M |
| Unique object terms | 47,345 | **586,369** |
| $\langle x_{i_R}, x_c, x_i \rangle$ (Keyword-based) | - | 11.4M |
| $\langle x_{i_R}, x_c, x_i \rangle$ (LLM-based) | 1M | 7.4M |
| $\langle x_{i_R}, x_c, x_i \rangle$ (Total) | 1M | **18.8M** |

Table 1: **Dataset statistics.** $\langle x_{t_R}, x_c, x_t \rangle$ denotes the triplet of *captions*, *i.e.*, {original caption, instruction, and modified caption}, and $\langle x_{i_R}, x_c, x_i \rangle$ denotes the *CIR triplet* of {original image, instruction, and modified image}.

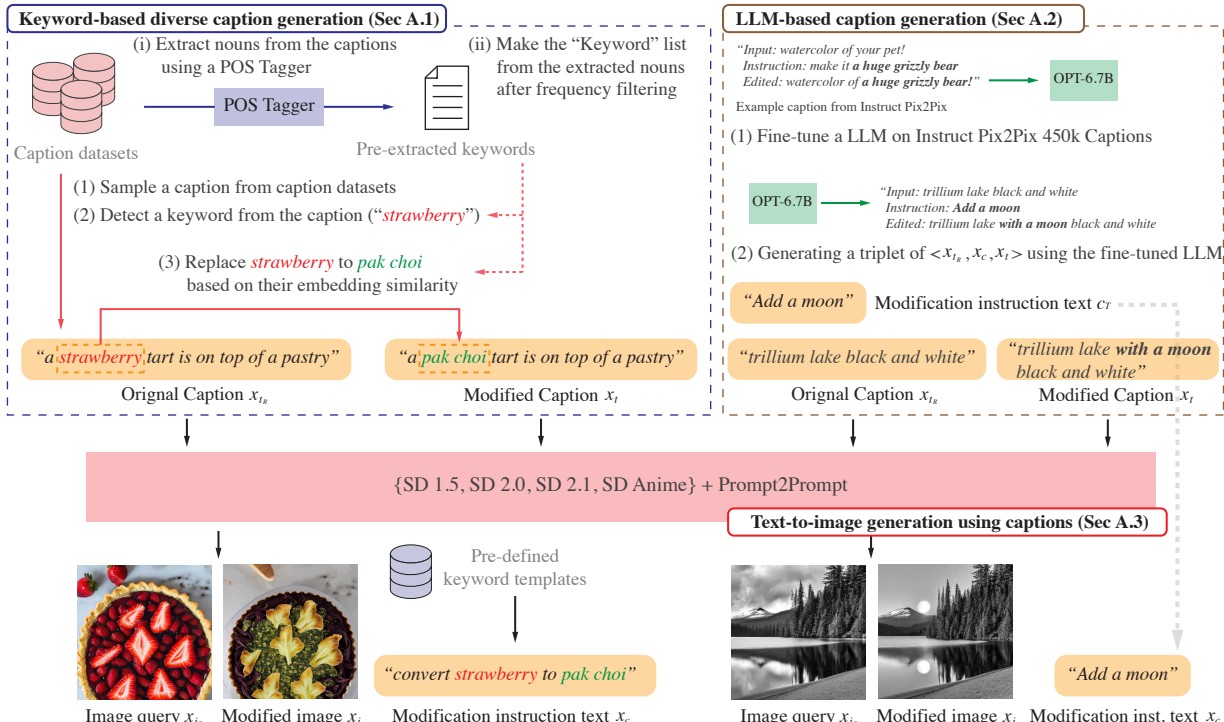

Figure 6: **Overview of the generation process for SynthTriplets18M.** $\langle x_{i_R}, x_c, x_i \rangle$ from $\langle x_{t_R}, x_c, x_t \rangle$.

ified captions by replacing the keywords in the reference caption (Section 4.1). (2) We fine-tune a large language model, OPT-6.7B (Zhang et al., 2022), on the generated caption triplets from Brooks et al. (2022) (Section 4.2). After generating massive triplets of $\langle x_{t_R}, x_c, x_t \rangle$, we generate images from the caption triplets using StableDiffusion (SD) and Prompt-to-Prompt Hertz et al. (2022) following IP2P (Section 4.3). We employ CLIP-based filtering to ensure high-quality triplets (Section 4.4). The entire generation process is illustrated in Fig. 6.

Compared to manual dataset collections (Wu et al., 2021; Liu et al., 2021), our approach can easily generate more diverse triplets even if a triplet rarely occurs in reality (See the examples in Fig. 6). For example, FashionIQ only contains text instructions for fashion items (*e.g.*, "is blue and has stripes"), but SynthTriplets18M contains more generic and various instructions. Furthermore, manually annotated instructions can be inconsistent and noisy due to the inherent noisiness of crowdsourcing. For example, CIRR has instructions with no information (*e.g.*, "same environment different species"), or instructions which ignore the original information (*e.g.*, "the target photo is of a lighter brown dog walking in white gravel along a wire and wooden fence"). On the other hand, SynthTriplets18M instructions only have modification information. Hence, a model trained on our CIR triplets can learn meaningful and generalizable representations, showing great ZS-CIR performances. Compared to the synthetic dataset of IP2P, our generation process is more scalable due to the keyword-based diverse caption generation process: Our caption triplets are synthesized based on keywords, SynthTriplets18M covers more diverse keywords than IP2P (47k vs. 586k as shown in Table 1). As a result, SynthTriplets18M contains more massive triplets (1M vs. 18M), and CIR models trained on SynthTriplets18M achieve better CIR performances even in the same dataset scale (1M).

## 4.1 Keyword-based diverse caption generation

As the first approach to generating caption triplets, we collect captions from the existing caption datasets and modify the captions by replacing the object terms in the captions, *e.g.*, ⟨"a strawberry tart is ...", "covert strawberry to pak choi", "a pak choi tart is ..."⟩ in Fig. 6. For the caption dataset, We use the captions from

| Templates to change ${source} to ${target} | | |
|---|---|---|
| *"replace ${source} with ${target}"* | *"substitute ${target} for ${source}"* | *"change ${source} to ${target}"* |
| *"${target}"* | *"${source} is removed and ${target} takes its place"* | *"alter ${source} to ${target}"* |
| *"apply ${target}"* | *"modify ${source} to become ${target}"* | *"swap ${source} for ${target}"* |
| *"convert ${source} to ${target}"* | *"customize ${source} to become ${target}"* | *"redesign ${source} as ${target}"* |
| *"replace ${source} with ${target}"* | *"change ${source} to match ${target}"* | *"turn ${source} into ${target}"* |
| *"update ${source} to ${target}"* | *"${target} is introduced after ${source} is removed"* | *"adapt ${source} to fit ${target}"* |
| *"substitute ${target} for ${source}"* | *"${target} is added in place of ${source}"* | *"choose ${target} instead"* |
| *"alter ${source} to match ${target}"* | *"${target} is introduced as the new option after"* | *"${target} is the new choice"* |
| *"upgrade ${source} to ${target}"* | *"${source} is removed and ${target} is added"* | *"${target} is the new selection"* |
| *"amend ${source} to fit ${target}"* | *"${source} is removed and ${target} is introduced"* | *"${target} is the new option"* |
| *"opt for ${target}"* | *"${target} is added as a replacement for ${source}"* | *"use ${target} from now on"* |
| *${source} is removed"* | *"${target} is the new option available"* | *"remodel ${source} into ${target}"* |
| *"add ${target}"* | *"${target} is added after ${source} is removed"* | *"revamp ${source} into ${target}"* |
| *"if it is ${target}"* | *"${target} is introduced after ${source} is retired"* | *"exchange ${source} with ${target}"* |
| *"${target} is the updated option"* | *"tweak ${source} to become ${target}"* | *"transform ${source} into ${target}"* |
| *"${target} is the updated choice"* | *"${source} is replaced with ${target}"* | *"${target} is the updated version"* |

Table 2: **The full 48 keyword converting templates.**

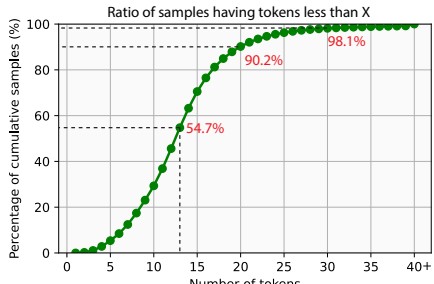

Figure 7: **Statistics of SynthTriplets18M instructions.** We show the population of our instruction captions $(z_{c_T})$ by the number of tokens per caption. We include captions having larger than 40 tokens in "40+".

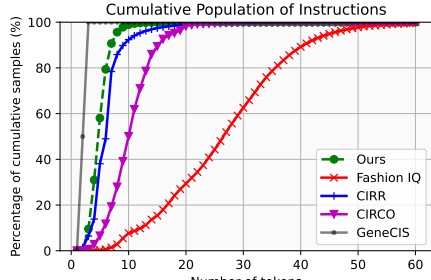

Figure 8: **Statistics of instructions of the CIR datasets.** We show the population of instruction captions (*e.g.*, "change A to B") by the number of tokens. We include captions having larger than 60 tokens in "60".

COYO 700M (Byeon et al., 2022), StableDiffusion Prompts[2] (user-generated prompts that make the quality of StableDiffusion better), LAION-2B-en-aesthetic (a subset of LAION-5B (Schuhmann et al., 2022a)) and LAION-COCO datasets (Schuhmann et al., 2022b) (synthetic captions for LAION-5B subsets with COCO style captions (Chen et al., 2015). LAION-COCO less uses proper nouns than the real web texts).

We extract the object terms from the captions using the part-of-speech (POS) tagger provided by `Spacy` [3]. After frequency filtering, we have 586k unique object terms (Table 1). To make the caption triplet $\langle x_{t_R}, x_c, x_t \rangle$, we replace the object term of each caption with other similar keywords by using the CLIP similarity score. More specifically, we extract the textual feature of keywords using the CLIP ViT-L/14 text encoder (Radford et al., 2021), and we choose an alternative keyword from keywords with a CLIP similarity between 0.5 and 0.7. By converting the original object to a similar object, we have caption pairs of $\langle x_{t_R}, x_t \rangle$.

Using the caption pair $\langle x_{t_R}, x_t \rangle$, we generate the modification instruction text $x_{c_T}$ based on a randomly chosen template from 48 pre-defined templates shown in Table 2. After this process, we have the triplet of $\langle x_{t_R}, x_c, x_t \rangle$. We generate ≈30M caption triplets by the keyword-based method.

## 4.2 Amplifying InstructPix2Pix (IP2P) triplets by LLM

We also re-use the generated $\langle x_{t_R}, x_c, x_t \rangle$ by IP2P. We amplify the number of IP2P triplets by applying the efficient LoRA fine-tuning (Hu et al., 2021) to OPT-6.7B (Zhang et al., 2022) on the generated 452k caption triplets provided by Brooks et al. (2022). Using the fine-tuned OPT, we generate ≈30M caption triplets.

### 4.3 Triplet generation from caption triplets

We generate 60M caption triplets $\langle x_{t_R}, x_c, x_t \rangle$ by the keyword-based generation process (Section 4.1) and the LLM-based generation process (Section 4.2) (See Section 4.5 for the statistics of the captions). We generate images for $x_{t_R}$ (original caption) and $x_t$ (modified caption) using state-of-the-art text-to-image generation models, such as StableDiffusion (SD) 1.5, 2.0, 2.1, and SD Anime. Following Brooks et al. (2022), we apply Prompt-to-Prompt (Hertz et al., 2022), which aims to generate similar images while keeping the identity of the original image (*e.g.*, the examples in Fig. 6). As a result, we generate 60M $\langle x_{i_R}, x_c, x_i \rangle$ ($z_{c_T}$ is given; $x_{i_R}$ and $x_i$ are generated by $x_{t_R}$ and $x_t$, respectively). While IP2P generates the samples only using SD 1.5, our generation process uses multiple DMs, for more diverse images not biased towards a specific model.

### 4.4 CLIP-based filtering

Our generation process can include low-quality triplets, *e.g.*, broken images or non-related image-text pairs. To prevent the issue, we apply a filtering process following Brooks et al. (2022) to remove the low-quality $\langle x_{i_R}, x_c, x_i \rangle$. First, we filter the generated images for an image-to-image CLIP threshold of 0.70 (between $x_{i_R}$ and $x_i$) to ensure that the images are not too different, an image-caption CLIP threshold of 0.2 to ensure that the images correspond to their captions (*i.e.*, between $x_{t_R}$ and $x_{i_R}$, and between $x_t$ and $x_i$), and a directional CLIP similarity (Gal et al., 2022) of 0.2 ($L_{\text{direction}} := 1 - \text{sim}(x_{i_R}, x_i) \cdot \text{sim}(x_{t_R}, x_t)$, where $\text{sim}(\cdot)$ is the CLIP similarity) to ensure that the change in before/after captions correspond with the change in before/after images. For keyword-based data generation, we filter out for a keyword-image CLIP threshold of 0.20 to ensure that images contain the keyword (*e.g.*, image-text CLIP similarity between the strawberry tart image and the keyword "strawberry" in Fig. 6). For instruction-based data generation, we filter out for an instruction-modified image CLIP threshold of 0.20 to ensure consistency with the given instructions.

The filtering process prevents low-quality triplets, such as broken images. For example, if a modified caption does not make sense to generate a corresponding image (*e.g.*, changing "apples are on the tree" to "apples are on the thunderstorm"), then StableDiffusion and Prompt2Prompt will not be able to generate proper images for the modified caption. These images will be filtered out by CLIP similarity because we measure the similarity between the generated images is sufficiently high, and the broken images will have low similarities with clean images. Similarly, if the original caption is not suitable for generating a corresponding image (*e.g.*, "invisible strawberry"), then it will fail to pass the keyword-image CLIP filtering.

After the filtering, we have 11.4M $\langle x_{i_R}, x_c, x_i \rangle$ from the keyword-based generated captions and 7.4M $\langle x_{i_R}, x_c, x_i \rangle$ from the LLM-based generated captions. It implies that the fidelity of our keyword-based method is higher than OPT fine-tuning in terms of T2I generation. As a result, SynthTriplets18M contains 18.8M synthetic $\langle x_{i_R}, x_c, x_i \rangle$. Examples of our dataset are shown in Fig. 9.

### 4.5 Dataset Statistics

We show the statistics of our generated caption dataset (*i.e.*, before T2I generation, $x_{t_R}$ and $x_t$). We use the CLIP tokenizer to measure the statistics of the captions. Fig. 7 shows the cumulative ratio of captions with tokens less than X. About half of the captions have less than 13 tokens, and 90% of the captions have less than 20 tokens. Only 0.8% of the captions have more than 40 tokens.

We also compare SynthTriplets18M, FashionIQ, CIRR, CIRCO, and GeneCIS in the token statistics of instructions (*i.e.*, $x_c$). Fig. 8 shows that the instruction statistics vary across different datasets. We presume that this is why the ZS-CIR still has difficulty outperforming the task-specific supervised CIR methods.

---

[2]https://huggingface.co/datasets/Gustavosta/Stable-Diffusion-Prompts
[3]https://spacy.io/

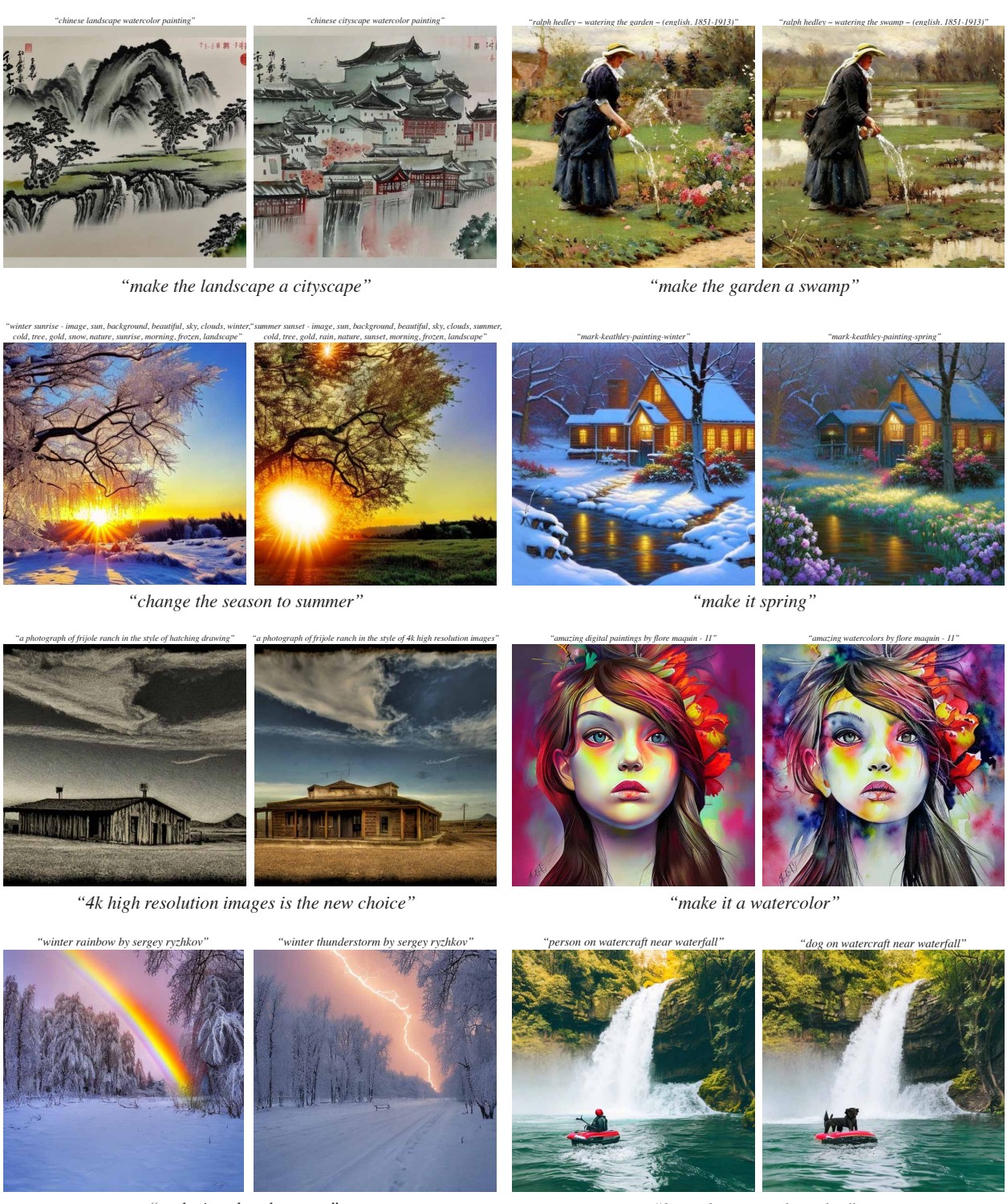

Figure 9: **Examples of SynthTriplets18M.** We show examples of $\langle x_{i_R}, x_c, x_i \rangle$, *i.e.*, {original image, modification instruction, and modified image}, as well as the generation prompt for $x_{i_R}$ and $x_i$.

## 5 Experiments

### 5.1 Implementation details

**Encoders.** We use three different CLIP models for image encoder (Fig. 4 "CLIP Img Enc"), the official CLIP ResNet-50 and ViT-L/14 (Radford et al., 2021), and CLIP ViT-G/14 by OpenCLIP (Ilharco et al., 2021), whose feature dimensions are 768, 1024, and 1280, respectively. Beyond the backbone size, we observe the choice of the text condition encoder is also important (Fig. 4 "Txt Enc"). As shown Balaji et al. (2022), using a text-oriented model such as T5 (Raffel et al., 2020) in addition to the CLIP textual encoder results in improved performance of text-to-image generation models. Motivated by this observation, we also use both the CLIP textual encoder and the language-oriented encoder for small image encoder (*i.e.*, CLIP ViT-L/14). We also observed the positive effect of the text-oriented model and experiment results showed that T5-XL, which has 3B parameters, could improve the performance by a large margin in the overall evaluation metrics.

**Denoiser.** We use a simple Transformer architecture for the denoising procedure, instead of the denoising U-Net (Rombach et al., 2022). We empirically observe that our transformer architecture performs slightly better than the U-Net architecture, but is much simpler. We use the multi-head self-attention blocks as the original Transformer (Vaswani et al., 2017). We set the depth, the number of heads, and the dimensionality of each head to 12, 16, and 64, respectively. The hidden dimension of the Transformer is set to 768 and 1280 for ViT-L and ViT-G, respectively. The denoising Transformer takes two inputs: a noisy visual embedding and a time-step embedding. The conditions (*e.g.*, text, mask and image conditions) are applied only to the cross-attention layer; thereby it is computationally efficient even using many conditions. CompoDiff is similar to the "DiT with cross-attention" by Peebles & Xie (2022), but handles more various conditions.

**Training details.** For the efficient training, all visual features are pre-extracted and frozen. All training text embeddings are extracted at every iteration. To improve computational efficiency, we reduced the number of input tokens of the T5 models to 77, as in CLIP. A single-layer perceptron was employed to align the dimension of text embeddings extracted from T5-XL with that of CLIP ViT-L/14.

### 5.2 Experiment settings

All models were trained using AdamW (Loshchilov & Hutter, 2017). We used DDIM (Song et al., 2020) for the sampling variance method. We did not apply any image augmentation but used pre-extracted CLIP image features for computational efficiency; text features were extracted on the fly as text conditions can vary in SynthTriplets18M. We report the detailed hyperparameters in Table A.1.

We evaluate the zero-shot (ZS) capability of CompoDiff on four CIR benchmarks, including FashionIQ (Wu et al., 2021), CIRR (Liu et al., 2021), CIRCO (Baldrati et al., 2023) and GeneCIS (Vaze et al., 2023). We compare CompoDiff to the recent ZS CIR methods, including Pic2Word (Saito et al., 2023) and SEARLE (Baldrati et al., 2023). We also reproduce the fusion-based methods, such as ARTEMIS (Delmas et al., 2022) and Combiner (Baldrati et al., 2022), on SynthTriplets18M and report their ZS performances. Note that the current CIR benchmarks are somewhat insufficient to evaluate the effectiveness of CompoDiff, particularly considering real-world CIR queries. More details are found in Section 5.4. Our work is the first study that shows the impact of the dataset scale and the zero-shot CIR performances with various methods, such as our method, ARTEMIS and Combiner. Due to the page limit, the details of each task and method are in Section 5.3 and 5.4. We also compare our training strategy with the other CIR methods in terms of the complexity in Section 5.3. In summary, we argue that our method is not specifically complex compared to the other methods. The fine-tuned performances on FashionIQ and CIRR are in Appendix B.

### 5.3 Comparison methods.

We compare CompoDiff with four state-of-the-art CIR methods as follows:

**Combiner** (Baldrati et al., 2022) involves a two-stage training process. First, the CLIP text encoder is fine-tuned by contrastive learning of $z_c$ and $z_{i_R} + z_c$ in the CLIP embedding space. The second stage replaces $z_{i_R} + z_c$ to the learnable Combiner module. Only the Combiner module is trained during the second stage.

**ARTEMIS** (Delmas et al., 2022) optimizes two similarities simultaneously. The implicit similarity is computed between the combined feature of $z_i$ and $z_c$, and the combined one of $z_{i_R}$ and $z_c$. The explicit matching is computed between $z_i$ and $z_c$. ARTEMIS suffers from the same drawback as previous CIR methods, *e.g.* TIRG (Vo et al., 2019): As it should compute combined feature of $z_i$ and $z_c$, it is not feasible to use an approximate nearest neighbor search algorithm, such as FAISS (Johnson et al., 2019). This is not a big problem in a small dataset like FashionIQ, but it makes ARTEMIS infeasible in real-world scenarios, *e.g.*, when the entire LAION-2B is the target database. As Combiner and ARTEMIS are trained based on CLIP ResNet-50 (RN50), we also train CompoDiff based on the same RN50 features for a fair comparison.

**Pic2Word** (Saito et al., 2023) projects a visual feature into text embedding space, instead of combining them. Pic2Word performs a text-to-image retrieval by using the concatenated feature as the input of the CLIP textual encoder. As the projection module is solely trained on cheap paired datasets without expensive triplet datasets, it is able to solve CIR in a zero-shot manner.

**SEARLE** (Baldrati et al., 2023) is a similar to Pic2Word, while SEARLE employs text-inversion task instead of projection. Baldrati et al. (2023) also proposed a technique, named Optimization-based Textual Inversion (OTI), a GPT-powered regularization method. In this paper, we compare SEARLE-XL and SEARLE-XL-OTI, which use the ViT-L/14 CLIP backbone for a fair comparison.

We also compare a naive editing-based retrieval method: editing an image using IP2P with an instruction and retrieving images using the CLIP image encoder. Note that Pic2Word and SEARLE are trained on the image-only datasets; therefore, it is impossible to train them on SynthTriplets18M. Similarly, Combiner and ARTEMIS only take triplets for training; hence, they cannot be trained with image-caption datasets, such as Conceptual Captions (Sharma et al., 2018) or LAION (Schuhmann et al., 2022a). Due to this reason, we only train the previous fusion-based methods, ARTEMIS and Combiner on SynthTriplets18M for comparison. Note that Combiner needs a pre-trained CLIP model trained on a large-scale image-caption dataset. In this point of view, we can argue that Combiner and CompoDiff are comparable in terms of the dataset scale; both methods use a large-scale image-caption dataset and synthetic triplets in our experiment.

### 5.4 CIR Datasets.

**FashionIQ** (Wu et al., 2021) has (46.6k / 15.5k / 15.5k) (training / validation / test) images with three fashion categories: Shirt, Dress, and Toptee. Each category has 18k training triplets and 12k evaluation triplets of $\langle x_{i_R}, x_c, x_i \rangle$. Examples of $x_c$ looks like: "is more strappy and emerald", "is brighter" or "is the same". The main drawback of FashionIQ is that the dataset is limited to the specific subsets of fashion domains, hence it is not possible to evaluate whether the methods are truly useful for real-world CIR tasks.

**CIRR** (Liu et al., 2021) contains more generic images than FashionIQ. CIRR uses the images from NLVR$^2$ (Suhr et al., 2018) with more complex and long descriptions. CIRR has 36k open-domain triplets divided into the train, validation, and test sets in 8:1:1 split. The examples of $x_c$ are "remove all but one dog and add a woman hugging it", "It's a full pizza unlike the other with the addition of spice bottles", or "Paint the counter behind the brown table white". As the example captions show, one of the main drawbacks of CIRR is that the text queries are not realistic compared to the real user scenario; we may need more shorter and direct instructions. As the text instruction distribution of CIRR is distinct from others (Fig. 8), we observe that sometimes CIRR results are a not reliable measure of open-world CIR tasks (*e.g.*, retrieval on LAION). Baldrati et al. (2023) also observe another main drawback of CIRR. CIRR contains a lot of false negatives (FNs) due to the nature of the image-text dataset (Chun et al., 2021; 2022; Chun, 2023)

**CIRCO.** To tackle the FN issue of CIRR, Baldrati et al. (2023) introduces the CIRCO dataset. This dataset comprises of 1020 queries, where 220 and 800 of them are used for validation and test, respectively. The images are from the COCO dataset (Lin et al., 2014), where the size of the image database is 120K COCO images. Example $x_c$ of CIRCO includes "has two children instead of cats" or "is on a track and has the front wheel in the air". We use mAP scores following Baldrati et al. (2023) which is known to be a robust retrieval metric (Musgrave et al., 2020; Chun et al., 2022).

**GeneCIS** (Vaze et al., 2023) is a dataset to evaluate different conditional similarities of four categories: (1) focus on an attribute, (2) change an attribute, (3) focus on an object, and (4) change an object. The first

| Method | Arch | Fashion IQ (Avg) | | CIRR | | CIRCO | | | GeneCIS |
| | | R@10 | R@50 | R@1 | $R_s$@1 | mAP@5 | mAP@10 | mAP@25 | R@1 |
|---|---|---|---|---|---|---|---|---|---|
| CLIP + IP2P[†] | ViT-L | 7.01 | 12.33 | 4.07 | 6.11 | 1.83 | 2.10 | 2.37 | 2.44 |
| Previous zero-shot methods (without SynthTriplets18M) | | | | | | | | | |
| Pic2Word[†] | ViT-L | 24.70 | 43.70 | 23.90 | 53.76 | 8.72 | 9.51 | 10.65 | 11.16 |
| SEARLE-OTI[†] | ViT-L | 27.51 | 47.90 | _24.87_ | 53.80 | 10.18 | 11.03 | 12.72 | - |
| SEARLE[†] | ViT-L | 25.56 | 46.23 | 24.24 | 53.76 | 11.68 | 12.73 | 14.33 | 12.31 |
| Zero-shot results with the models trained with SynthTriplets18M | | | | | | | | | |
| ARTEMIS | RN50 | 33.24 | 47.99 | 12.75 | 21.95 | 9.35 | 11.41 | 13.01 | 13.52 |
| Combiner | RN50 | 34.30 | 49.38 | 12.82 | 24.12 | 9.77 | 12.08 | 13.58 | 14.93 |
| CompoDiff | RN50 | 35.62 | 48.45 | 18.02 | 57.16 | 12.01 | 13.28 | 15.41 | 14.65 |
| CompoDiff | ViT-L | 36.02 | 48.64 | 18.24 | 57.42 | 12.55 | 13.36 | _15.83_ | 14.88 |
| CompoDiff | ViT-L & T5-XL | _37.36_ | _50.85_ | 19.37 | _59.13_ | _12.31_ | _13.51_ | 15.67 | _15.11_ |
| CompoDiff | ViT-G | **39.02** | **51.71** | **26.71** | **64.54** | **15.33** | **17.71** | **19.45** | **15.48** |

Table 3: **Zero-shot CIR comparisons.** † denotes the results by the official model weight, otherwise, models are trained on SynthTriplets18M and LAION-2B (ARTEMIS and Combiner are trained solely on SynthTriplets18M, while CompoDiff is trained on both). ‡ refers to the use of both the CLIP textual encoder and T5-XL as the encoder for the text condition. The full results for each dataset are shown in Appendix B.

two categories are based on the VAW dataset (Pham et al., 2021), which contains massive visual attributes in the wild. The other categories are sampled from COCO.

**The limitations of the current CIR benchmarks.** One of the significant drawbacks of the existing CIR benchmarks, such as FashionIQ (Wu et al., 2021) and CIRR (Liu et al., 2021), is the domain-specific characteristics that cannot be solved without training on the datasets. For example, the real-world queries that we examine are mainly focused on small editing, addition, deletion, or replacement. However, because the datasets are constructed by letting human annotators write a modification caption for a given two images, the text conditions are somewhat different from the real-world CIR queries. For example, CIRR dev set has some text conditions like: "show three bottles of soft drink" (different from the common real-world CIR text conditions), "same environment different species" (ambiguous condition), "Change the type of dog and have it walking to the left in dirt with a leash." (multiple conditions at the same time). These types of text conditions are extremely difficult to solve in a zero-shot manner, but we need access to the CIRR training dataset. Instead, we believe that CIRCO is a better benchmark than FashionIQ and CIRR. It is because CIRCO aims to resolve the false negatives of FashionIQ and CIRR, focusing on the retrieval evaluation.

Furthermore, the existing benchmarks (even more recent ones, such as CIRCO and GeneCIS) cannot evaluate the negative and mask conditions. In practice, when we perform a qualitative study on a large image index (*i.e.*, LAION-2B image index), we observe that CompoDiff outperforms previous methods, such as Pic2Word, in terms of the retrieval quality (See Fig. 11 and Fig. C.1 for examples). Also, we observe that even if the CIR scores are similar (*e.g.*, 10M and 18.8M in Table 6), in the LAION-2B index, the retrieval quality can be significantly better. Unfortunately, it is impossible to evaluate the quality of retrieval results on the LAION-2B, because a quantitative study requires expensive and infeasible human verification.

### 5.5 Qualitative comparisons on four Zero-shot CIR (ZS-CIR) benchmarks

Table 3 shows the overview of ZS-CIR comparison results. CLIP + IP2P denotes the naive editing-based approach by editing the reference image with the text condition using IP2P and performing image-to-image retrieval using CLIP ViT-L. In the table, CompoDiff outperforms all the existing methods with significant gaps. The table shows the effectiveness both of our diffusion-based CIR approach and our massive synthetic dataset. In the SynthTriplets18M-trained group, CompoDiff outperforms previous SOTA fusion-based CIR methods with a large gap, especially on CIRR and CIRCO, which focus on real-life images and complex descriptions. Our improvement is not main due to the architecture, as CompoDiff already outperforms the fusion methods in RN50. We also can observe that the SynthTriplets18M-trained group also enables the

fusion-based methods to have the zero-shot capability competitive to the SOTA zero-shot CIR methods, Pic2Word and SEARLE. More interestingly, ARTEMIS even outperforms its supervised counterpart on FashionIQ with a large gap (40.6 vs. 38.2 in average recall). We provide the full results of Table 3, the supervised results of ARTEMIS, Combiner, and CompoDiff trained on SynthTriplets18M in Appendix B.

Compared to the previous ZS-CIR methods (Pic2Word and SEARLE), CompoDiff achieves remarkable improvements on the same architecture scale (*i.e.*, ViT-L), except on CIRR. We argue that it is due to the noiseness of the CIRR dataset. Instead, CompoDiff outperforms the other methods on FashionIQ, CIRCO and GeneCIS with a significant gap. We believe that it is because CompoDiff explicitly utilizes the diverse and massive synthetic triplets, while Pic2Word and SEARLE only employ images and the "a photo of" caption during training, resulting in a lack of diversity and generalizability.

## 5.6 Inference time

One possible drawback of CompoDiff is a relatively slow inference compared to encoder-only methods. However, we confirm that CompoDiff is practically useful with high throughput ($120 \sim 230$ms) for a single image (Table 4). The table reports throuputs of the comparison methods on ViT-L/14 backbone (Pic2Word, SEARLE, CompoDiff) or RN50 backbone (ARTEMIS, Combiner). One of the advantages of CompoDiff is that we can control the trade-off between retrieval performance and inference time. Note that it is impossible for the other methods. If we need a faster inference time, even with a worse retrieval performance, we can reduce the number of diffusion steps as shown in Table 4. Even with only 4 or 5 iterations, our model can produce competitive results. If we use 100 steps, we have a slightly better performance (42.65 vs. 42.17), but a much slower inference time (2.02 sec vs. 0.12 sec). In the experiments, we set the step size to 10.

| Step | 1 | 2 | 3 | 4 | 5 | 10 | 50 | 100 | Pic2Word | SEARLE | ARTEMIS | Combiner |
|---|---|---|---|---|---|---|---|---|---|---|---|---|
| Avg(R@10, R@50) | 21.52 | 29.22 | 37.63 | **41.08** | **42.17** | **42.33** | 42.45 | 42.65 | 34.20 | 35.90 | 40.61 | 41.84 |
| Time (sec) | 0.02 | 0.05 | 0.07 | **0.09** | **0.12** | **0.23** | 1.08 | 2.02 | 0.02 | 0.02 | 0.005 | 0.006 |
| GPU Memory (bs 1) | | | | 2.60 GB | | | | | 0.97 GB | 1.32 GB | 0.26 GB | 0.54 GB |
| GPU Memory (bs 256) | | | | 5.80 GB | | | | | 2.99 GB | 3.24 GB | 2.82 GB | 1.95 GB |

Table 4: **Performances vs. inference time by varying the number of denoising steps.** Numbers are measured on the FashionIQ validation split. CompoDiff (ViT-L) in Table 3 is equivalent to 10 steps, but using 5 steps is a practical alternative. The inference time was measured on a single A100 GPU with a batch size of 1. GPU memories are the pick-allocated memory measured on batch sizes of 1 and 64, respectively. Note that ARTEMIS cannot support efficient batch operation for inference because its forward path needs triplet information, not an image-instruction pair; here, we report ARTEMIS information with training triplets.

Our another contribution is a novel and efficient conditioning for the diffusion model. Instead of using a concatenated vector of all conditions and inputs as the input of the diffusion model, we use the cross-attention mechanism for conditions and leave the input size the same as the original size. As shown in the next section (Table 5), our design choice is three times faster than the naive implementation.

We also remark that recently, there have been huge advances in boosting diffusion model inference speed, such as the Consistency Model (Song et al., 2023) or distillation (Meng et al., 2023). Using more efficient diffusion model variants for the CompoDiff model will also greatly improve the inference speed. However, these approaches will need a completely different training framework (Song et al., 2023) or a complicated two-staged distillation process (Meng et al., 2023). While we believe that applying these techniques to CompoDiff will bring a huge benefit, we leave further improvements for future work.

## 5.7 Analysis

**Denoising Transformer ablation.** CompoDiff does not take textual embeddings $z_{c_T}$ as concatenated input tokens of the denoising Transformer but as a condition of cross-attention for the efficient inference. We compare the impact of different design choices for handling textual embeddings. First, we evaluate the Dall-e2 "Prior" model (Ramesh et al., 2022) which converts CLIP textual embeddings into CLIP visual embeddings (we use a public community model (Lee et al., 2022) because the official model is not yet

| Method | COCO 5k | | | CxC | | | ECCV Caption | | | Throughput |
| | R@1 | R@5 | R@10 | R@1 | R@5 | R@10 | mAP@R | R-P | R@1 | images/sec |
|---|---|---|---|---|---|---|---|---|---|---|
| Image to text retrieval | | | | | | | | | | |
| Prior (Lee et al., 2022) | 32.04 | 56.84 | 67.68 | 33.76 | 60.50 | 71.32 | 14.19 | 23.37 | 46.47 | 497.28 |
| Prior-like Stage1 | 34.32 | 58.40 | 69.52 | 35.01 | 62.35 | 74.21 | 16.35 | 25.20 | 49.01 | 497.28 |
| Ours (Stage 1 only) | 35.13 | 59.46 | 70.26 | 35.30 | 62.62 | 74.02 | 16.01 | 25.44 | 49.64 | 1475.52 |
| Ours (Stage 1 + Stage2) | 33.20 | 58.00 | 68.94 | 34.78 | 61.68 | 72.96 | 15.07 | 24.39 | 47.03 | 1473.92 |
| Text to image retrieval | | | | | | | | | | |
| Prior (Lee et al., 2022) | 17.05 | 34.25 | 43.16 | 18.62 | 37.43 | 47.07 | 18.10 | 26.46 | 46.40 | 497.28 |
| Prior-like Stage1 | 22.62 | 38.31 | 48.11 | 21.42 | 41.42 | 51.79 | 20.70 | 29.80 | 51.50 | 497.28 |
| Ours (Stage 1 only) | 22.47 | 39.18 | 49.08 | 22.51 | 42.40 | 52.77 | 21.46 | 30.30 | 53.75 | 1475.52 |
| Ours (Stage 1 + Stage2) | 20.00 | 38.63 | 48.25 | 21.57 | 41.71 | 51.99 | 20.82 | 29.84 | 51.65 | 1473.92 |

Table 5: **Comparisons of design choices for handling textual embeddings on cross-modal retrieval benchmarks.** Throughput was measured on 1 A100 GPU with a batch size of 32. For all metrics, higher is better. Here, we use the community version of the prior model by Lee et al. (2022) because the official prior model (Ramesh et al., 2022) is not public.

publically available). Second, we test the "Prior-like" model by using the denoising Transformer, but taking text guidance as input tokens instead of cross-attention. We also test two more CompoDiff models from our two-stage training strategy.

As the Prior models are not designed for handling CIR triplets, we measure their ability on image-to-text (I2T) and text-to-image (T2I) cross-modal retrieval benchmarks on the MS-COCO Caption dataset (Chen et al., 2015). We also evaluate them on the extension of COCO Caption to mitigate the false negative problem of COCO, CxC (Parekh et al., 2020) and ECCV Caption (Chun et al., 2022). Table 5 shows the average I2T and T2I metrics of each benchmark. In the table, we first observe that our design choice is three times faster than the "Prior-ish" counterparts by handling textual embeddings with cross-attention. Second, we observe that Stage 1 only CompoDiff shows a better understanding of I2T and T2I tasks. We speculate that this is because Ours (Stage 1 only) is directly optimized by the image-to-text (ITM) matching style dataset, while Ours (Stage 1 + Stage 2) is also trained with other types of conditions (*e.g.*, masks, negative texts, image conditions). In summary, our design choice shows $\times$ 3 faster inference time than the prior model (Ramesh et al., 2022) but better cross-modal retrieval performances on COCO and its extensions.

**Impact of dataset scale.** Table 6 shows the impact of the dataset scale by SynthTriplets18M on ARTEMIS, Combiner and CompoDiff. First, at a scale of 1M, models trained on our 1M subset significantly outperformed the IP2P triplets. This result indicates that our dataset has a more diverse representation capability. As the size of our dataset increases, the performance gradually improves. Notably, SynthTriplets18M shows consistent performance improvements from 1M to 18.8M, where manually collecting triplets in this scale is infeasible and nontrivial. Thanks to our diversification strategy, particularly keyword-based generation, we can scale up the triplet to 18.8M without manual human labor.

| | IP2P(1M) | 1M | 5M | 10M | 18.8M |
|---|---|---|---|---|---|
| FashionIQ Avg(R@10, R@50) | | | | | |
| ARTEMIS | 26.03 | 27.44 | 36.17 | 41.35 | 40.62 |
| Combiner | 29.83 | 29.64 | 35.23 | 41.81 | 41.84 |
| CompoDiff | 27.24 | 31.91 | 38.11 | 42.41 | 42.33 |
| CIRR Avg(R@1, $R_s$@1) | | | | | |
| ARTEMIS | 14.91 | 15.12 | 15.84 | 17.56 | 17.35 |
| Combiner | 16.50 | 16.88 | 17.21 | 18.77 | 18.47 |
| CompoDiff | 27.42 | 28.32 | 31.50 | 37.25 | 37.83 |

Table 6: **Impact of dataset scale.** IP2P denotes the public 1M synthetic dataset by (Brooks et al., 2022).

Table 6 shows that the massive data points are not the necessary condition for training CompoDiff, but all methods are consistently improved by scaling up the data points. It is also worth noting that although the FashionIQ and CIRR scores look somewhat saturated after 10M, these scores cannot represent authentic CIR performances due to the limitations of the datasets as discussed in Section 5.4. As far as we know, this is the first study shows the impact of the dataset scale to the zero-shot CIR performances.

**Condition text encoder.** As observed by Balaji et al. (2022) and Byun et al. (2024), the power of the CLIP textual encoder affects a lot to the performance of image-text tasks. Motivated by this observation, we also use both the CLIP textual encoder and the language-oriented encoder for extracting the text features of Eq. (3). Table 7 shows the choice of the

| Text Enc | T5-XL | CLIP | CLIP + T5-XL |
|---|---|---|---|
| FashionIQ | 38.20 | 42.33 | **44.11** |
| CIRR | 29.19 | 37.83 | **39.25** |

Table 7: **Impact of text encoder.**

text encoder affects a lot to the performance on ViT-L/14 CLIP backbone. When the CLIP textual encoder and the T5-XL were used together, the results improved significantly. We suspect that this is because the strong T5 encoder can help the CLIP text encoder to better understand given captions. Interestingly, we observe that using T5 alone degrades the performance even compared to using the CLIP textual encoder alone. We presume this is because T5-XL is specified for long text sequences (*e.g.*, larger than 100 tokens) and text-only data. Meanwhile, SynthTriplets18M has a short average length (See Fig. 7 and 8), which is not specialized by T5. Also, our dataset is based on captions, paired with an image; we also need to consider image information to understand the given caption, but we cannot handle image information alone with T5.

Therefore, we use both T5-XL encoder (Raffel et al., 2020) and CLIP text encoder for the ViT-L model. For the ViT-G model, we use the CLIP text encoder only because we empirically observe that the textual representation power of the ViT-G is much more powerful than the ViT-L. Exploring the power of textual representations, such as learnable prompts to the CFG guidance, could be an interesting future work.

**Stage 2 ablation.** As we described in Section 3.2, we alternatively update the model using three different objectives in Stage 2. Here, we conduct an ablation study of our design choice in Table 8. Our alternative learning strategy improves the overall performance. It is because although SynthTriplets18M is a vast and diverse dataset, its diversity is weaker than LAION. While we train CompoDiff to handle triplets by training on SynthTriplets18M using Eq. (3), we employ additional tasks using LAION, *i.e.*, Eq. (1) and Eq. (2) for achieving a better generalizability.

| | *only* Eq. (3) | Proposed |
|---|---|---|
| FIQ | 41.15 | **42.33** |
| CIRR | 35.25 | **37.83** |

Table 8: **Stage 2 ablation.** Compared by FashionIQ Avg(R@10, R@50) and CIRR Avg(R@1 R_s@1).

**The choice of $w_I$ and $w_T$.** We include the retrieval performances by varying conditions in Fig. 10. As $w_I$ increases, the generated image embeddings become more dependent on the reference image, while increasing $w_T$ results in a greater influence of the text guidance (See Fig. 5). However, large $w_I$ and $w_T$ are not always beneficial. If $w_I$ or $w_T$ is too large, it can lead to unexpected results. To find a harmonious combination of $w_I$ and $w_T$, we performed a sweeping process as shown in Fig. 10. We use $w_I$ as 1.5 and $w_T$ as 7.5 considering the best content-condition trade-off.

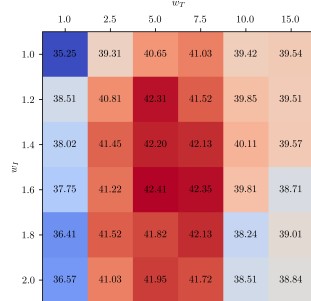

Figure 10: $w_I$ **and** $w_T$ **vs. FashionIQ ZS-CIR performance.**

## 5.8 Qualitative examples

We qualitatively show the versatility of CompoDiff for handling various conditions. For example, CompoDiff not only can handle a text condition, but it can also handle a *negative* text condition (*e.g.*, removing specific objects or patterns in the retrieval results), masked text condition (*e.g.*, specifying the area for applying the text condition). CompoDiff even can handle all conditions simultaneously (*e.g.*, handling positive and negative text conditions with a partly masked reference image at the same time). To show the quality of the retrieval results, we conduct a zero-shot CIR on entire LAION dataset (Schuhmann et al., 2022a) using FAISS (Johnson et al., 2019) for simulating billion-scale CIR scenarios.

Fig. 11 shows qualitative comparsions of zero-shot CIR results by Pic2Word and CompoDiff. CompoDiff results in semantically high-quality retrieval results (*e.g.*, understanding the "crowdedness" of the query image and the meaning of the query text at the same time). However, Pic2Word shows poor understanding of the given queries, resulting in unfortunate retrieval results (*e.g.*, ignoring "grown up" of text query, or the "crowdedness" of the query image). More examples are in Fig. C.1.

Finally, it is worth noting that CompoDiff generates a feature belonging to the CLIP visual latent space. Namely, unCLIP (Ramesh et al., 2022), which decodes a CLIP image feature to an image, can be applied

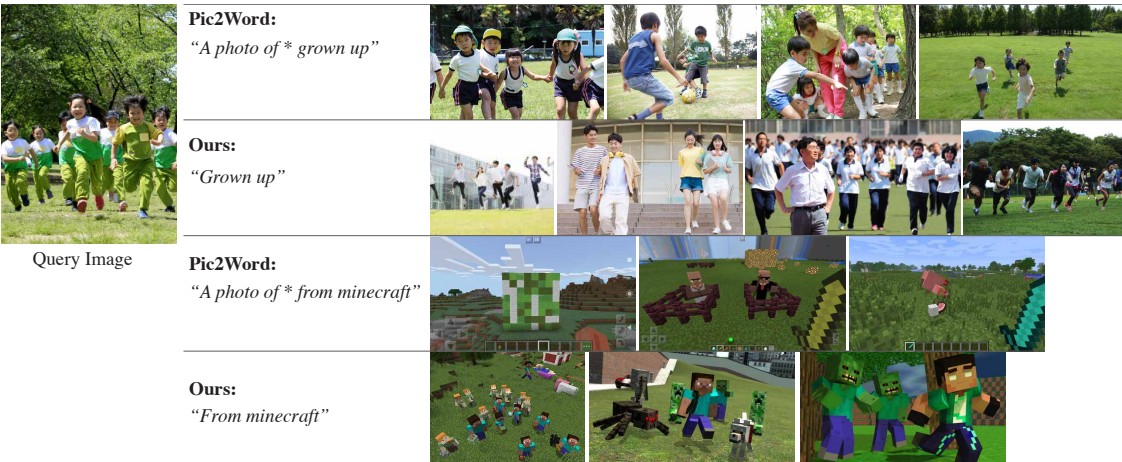

Figure 11: **Qualitative comparison of zero-shot CIR for Pic2Word and CompoDiff.** We conduct CIR on LAION. As Pic2Word cannot take a simple instruction, we made a simple modification for the given instruction.

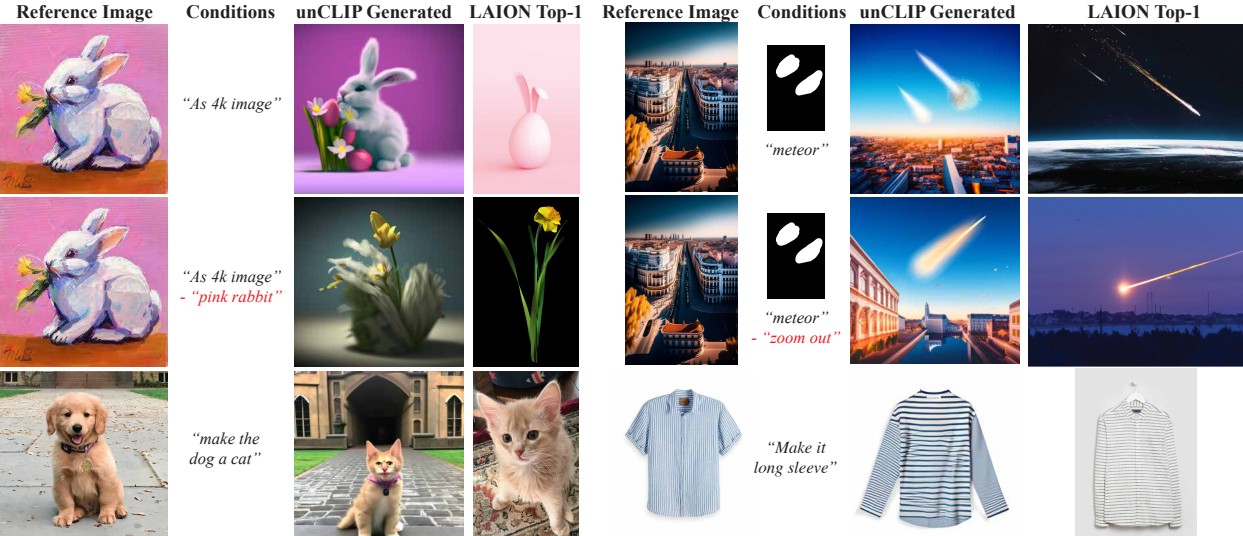

Figure 12: **Generated and retrieved images by CompoDiff.** Images are generated by unCLIP decoder and retrieved from LAION using transformed features by CompoDiff. More examples are shown in Appendix C.

to our composed features. We compare the top-1 retrieval results from LAION and the generated images in Fig. 12 and Appendix C. We use the Karlo unCLIP decoder (Lee et al., 2022), by replacing the original Prior module to CompoDiff. As shown in the figures, CompoDiff can manipulate the given input reflecting the given conditions. We believe that incorporating unCLIP into the real-world search engine could potentially improve the user experience by generating images when the desired search results are not available.

## 6   Conclusion

We have introduced CompoDiff, a novel diffusion-based method for solving complex CIR tasks. We have created a large and diverse dataset named SynthTriplets18M, consisting of 18.8M triplets of images, modification texts, and modified images. CompoDiff has demonstrated impressive ZS-CIR capabilities, as well as remarkable versatility in handling diverse conditions, such as negative text or image masks, and the controllability to enhance user experience, such as adjusting image text query weights. Furthermore, by training the existing CIR methods on SynthTriplets18M, the models became comparable ZS predictors to the ZS-CIR methods. We strongly encourage future researchers to leverage our dataset to advance the field of CIR.

## Societal Impact

Our work is primarily focused on solving complex composed image retrieval (CIR) challenges and is not designed for image editing purposes. However, we are aware that with the use of additional public resources, such as the community version of the unCLIP feature decoder (Ramesh et al., 2022), our method can potentially be utilized as an image editing method. We would like to emphasize that this unintended application is not the primary objective of our research, and we cannot guarantee the effectiveness or safety of our method in this context. It is important to note that we have taken steps to mitigate potential risks associated with the unintended use of our method for image editing. For instance, we applied NSFW filters to filter out potentially malicious samples during the creation of SynthTriplets18M. Nevertheless, we recognize the need for continued research into the ethical and societal implications of AI technologies and pledge to remain vigilant about potential unintended consequences of our work.

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

## Appendix

In this additional document, we provide the detailed training hyperparameter settings (Appendix A), the full experimental results (Appendix B) and more qualitative examples (Appendix C).

## A Hyperparameter details

Table A.1 shows the detailed training hyperparameters of CompoDiff.

|  | Stage1 | Stage2 | Fine-tuning |
|---|---|---|---|
| Diffusion steps | 1000 | 1000 | 1000 |
| Noise schedule | cosine | cosine | cosine |
| Sampling steps | 10 | 10 | 10 |
| Sampling variance method | DDIM | DDIM | DDIM |
| Dropout | 0.1 | 0.1 | 0.1 |
| Weight decay | 6.0e-2 | 6.0e-2 | 6.0e-2 |
| Batch size | 4096 | 2048 | 2048 |
| Iterations | 1M | 200K | 50K |
| Learning rate | 1e-4 | 1e-5 | 1e-5 |
| Optimizer | AdamW | AdamW | AdamW |
| EMA decay | 0.9999 | 0.9999 | 0.9999 |
| Input tokens | $z_i^{(t)}$, t | $z_i^{(t)}$, t | $z_i^{(t)}$, t |
| Conditions | $z_{c_T}$ | $z_{c_T}$, $z_{i_R}$, $z_{c_M}$ | $z_{c_T}$, $z_{i_R}$ |
| Training dataset | LAION-2B | LAION-2B, SynthTriplets18M | FashionIQ or CIRR trainset |
| Image encoder | CLIP-L/14 | CLIP-L/14 | CLIP-L/14 |
| Text encoder | CLIP-L/14 | CLIP-L/14 | CLIP-L/14 |
| Denoiser depth | 12 | 12 | 12 |
| Denoiser heads | 16 | 16 | 16 |
| Denoiser head channels | 64 | 64 | 64 |

Table A.1: **Hyperparameters.** A model trained by Stage 1 and Stage 2 is equivalent to "Zero-shot" in the main table. A "supervised model" is the same as the fine-tuned version.

## B Full experiment results of Table 3

In this subsection, we report the full experiment results of Table 3. For FashionIQ and CIRR datasets, we also report additional fine-tuning results of SynthTriplets18M-trained CIR methods (*i.e.*, directly fine-tune the "zero-shot" methods on the target training dataset).

**FashionIQ.** Table B.1 shows the comparison of CompoDiff with baselines on the FashionIQ dataset. Following the standard choice, we use recall@K as the evaluation metric. "Zero-shot" means that the models are not trained on FashionIQ (*i.e.*, same as Table 3). ARTEMIS and Combiner were originally designed for the supervised setting, but, we trained them on SynthTriplets18M for a fair comparison with our method. Namely, we solely train them on SynthTriplets18M for the zero-shot benchmark and fine-tune the zero-shot weights on the FashionIQ training set for the supervised benchmark.

**CIRR.** Table B.2 shows the CIRR results and all experimental settings were identical to FashionIQ. Similar to FashionIQ, CompoDiff also achieves a new state-of-the-art CIRR zero-shot performance. It is noteworthy that Combiner performs great in the supervised setting but performs worse than CompoDiff in the zero-shot setting. We presume that the fine-tuned Combiner text encoder is overfitted to long-tailed CIRR captions. It is partially supported by our additional experiments on text encoder in Section 5.7; a better understanding of complex texts provides better performances.

For both datasets, we achieve the best scores by fine-tuning the Combiner model trained on SynthTriplets18M to the target dataset. It shows the benefits of our dataset compared to the limited CIR triplet datasets.

| | Shirt | | Dress | | Toptee | | Average | | |
|---|---|---|---|---|---|---|---|---|---|
| Method | R@10 | R@50 | R@10 | R@50 | R@10 | R@50 | R@10 | R@50 | Avg. |
| Previous zero-shot methods (without SynthTriplets18M) | | | | | | | | | |
| CLIP + InstructPix2Pix | 7.24 | 12.71 | 6.31 | 10.42 | 7.49 | 13.85 | 7.01 | 12.33 | 9.67 |
| Pic2Word | 26.20 | 43.60 | 20.00 | 40.20 | 27.90 | 47.40 | 24.70 | 43.70 | 34.20 |
| SEARLE-XL-OTI | 30.37 | 47.49 | 21.57 | 44.47 | 30.90 | 51.76 | 27.61 | 47.90 | |
| SEARLE-XL | 26.89 | 45.58 | 20.48 | 43.13 | 29.32 | 49.97 | 25.56 | 46.23 | |
| Zero-shot results with the models trained with SynthTriplets18M | | | | | | | | | |
| ARTEMIS[†] | 30.70 | 50.43 | 33.52 | 46.54 | 35.49 | 47.01 | 33.24 | 47.99 | 40.62 |
| CLIP4Cir[†] | 32.32 | 51.65 | 34.92 | 48.38 | 35.65 | 48.10 | 34.30 | 49.38 | 41.84 |
| CompoDiff (ViT-L)[†] | 37.69 | 49.08 | 32.24 | 46.27 | 38.12 | 50.57 | 36.02 | 48.64 | 42.33 |
| CompoDiff [‡] (ViT-L)[†] | 38.10 | 52.48 | 33.91 | 47.85 | 40.07 | 52.22 | 37.36 | 50.85 | 44.11 |
| CompoDiff (ViT-G)[†] | **41.31** | **55.17** | **37.78** | **49.10** | **44.26** | **56.41** | **39.02** | **51.71** | **46.85** |
| Supervised | | | | | | | | | |
| JVSM | 12.00 | 27.10 | 10.70 | 25.90 | 13.00 | 26.90 | 11.90 | 26.60 | 19.25 |
| CIRPLANT w/ OSCAR | 17.53 | 38.81 | 17.45 | 40.41 | 21.64 | 45.38 | 18.87 | 41.53 | 30.20 |
| TRACE w/ BERT | 20.80 | 40.80 | 22.70 | 44.91 | 24.22 | 49.80 | 22.57 | 46.19 | 34.38 |
| VAL w/ GloVe | 22.38 | 44.15 | 22.53 | 44.00 | 27.53 | 51.68 | 24.15 | 46.61 | 35.38 |
| MAAF | 21.30 | 44.20 | 23.80 | 48.60 | 27.90 | 53.60 | 24.30 | 48.80 | 36.55 |
| ARTEMIS | 21.78 | 43.64 | 27.16 | 52.40 | 29.20 | 54.83 | 26.05 | 50.29 | 38.17 |
| CurlingNet | 21.45 | 44.56 | 26.15 | 53.24 | 30.12 | 55.23 | 25.90 | 51.01 | 38.46 |
| CoSMo | 24.90 | 49.18 | 25.64 | 50.30 | 29.21 | 57.46 | 26.58 | 52.31 | 39.45 |
| RTIC-GCN w/ GloVe | 23.79 | 47.25 | 29.15 | 54.04 | 31.61 | 57.98 | 28.18 | 53.09 | 40.64 |
| DCNet | 23.95 | 47.30 | 28.95 | 56.07 | 30.44 | 58.29 | 27.78 | 53.89 | 40.84 |
| AACL | 24.82 | 48.85 | 29.89 | 55.85 | 30.88 | 56.85 | 28.53 | 53.85 | 41.19 |
| SAC w/ BERT | 28.02 | 51.86 | 26.52 | 51.01 | 32.70 | 61.23 | 29.08 | 54.70 | 41.89 |
| MUR | 30.60 | 57.46 | 31.54 | 58.29 | 37.37 | 68.41 | 33.17 | 61.39 | 47.28 |
| Combiner | 39.99 | 60.45 | 33.81 | 59.40 | 41.41 | 65.37 | 38.32 | 61.74 | 50.03 |
| Pre-training with SynthTriplets18M & Fine-tuning with FashionIQ | | | | | | | | | |
| ARTEMIS[†] | 32.17 | 53.32 | 34.80 | 48.10 | 36.58 | 47.63 | 34.52 | 49.68 | 42.10 |
| Combiner[†] | 37.21 | **60.71** | **42.75** | **60.50** | 42.98 | **65.49** | **40.98** | **62.23** | **51.61** |
| CompoDiff (ViT-L)[†] | 40.88 | 53.06 | 35.53 | 49.56 | 41.15 | 54.12 | 39.05 | 52.34 | 46.31 |
| CompoDiff (ViT-L + T5-XL)[†] | 40.65 | 57.14 | 36.87 | 57.39 | 43.93 | 61.17 | 40.48 | 58.57 | 49.53 |
| CompoDiff (ViT-G)[†] | **41.68** | 56.02 | 38.39 | 51.03 | **45.70** | 57.32 | 39.81 | 51.90 | 47.73 |

Table B.1: **Comparisons on FashionIQ.** The "Zero-shot" scenario performs CIR using a model not trained on the FashionIQ dataset, while models are trained on FashionIQ for the "Supervised" scenario. The last group denotes that a model is trained on SynthTriplets18M. † denotes that the models are newly trained by us.

**CIRCO.** The detailed CIRCO results are shown in Table B.3. In the table, we can observe that in all metrics, CompoDiff achieves the best performances among the zero-shot CIR methods. This result supports the effectiveness of CompoDiff and SynthTriplets18M in real-world CIR tasks.

**GeneCIS.** Finally, we report detailed GeneCIS in Table B.4. In the table, CompoDiff shows the best performance in the average recall. Our method especially outperforms the other methods in "Change Attribute", "Focus Object" and "Change Object". On the other hand, CompoDiff is less effective than Pic2Word and SEARLE in "Focus attribute". We presume that it is because the instruction distribution of "Focus Attribute" differs a lot from the instruction of SynthTriplets18M. Among other CIR methods in the SynthTriplets18M-trained group, CompoDiff shows the best performances.

| Method | R@1 | R@5 | R@10 | R@50 | $R_s$@1 | $R_s$@2 | $R_s$@3 | Avg(R@1, $R_s$@1) |
|---|---|---|---|---|---|---|---|---|
| Previous zero-shot methods (without SynthTriplets18M) | | | | | | | | |
| CLIP + InstructPix2Pix | 4.07 | 8.41 | 11.2 | 15.38 | 6.11 | 10.05 | 13.33 | 5.09 |
| Pic2Word | 23.90 | 51.70 | 65.30 | 87.80 | 53.76 | 74.46 | 87.08 | 38.83 |
| SEARLE-XL-OTI | 24.87 | 52.31 | 66.29 | 88.58 | 53.80 | 74.31 | 86.94 | 39.34 |
| SEARLE-XL | 24.24 | 52.48 | 66.29 | 88.84 | 53.76 | 75.01 | 88.19 | 39.00 |
| Zero-shot results with the models trained with SynthTriplets18M | | | | | | | | |
| ARTEMIS[†] | 12.75 | 33.84 | 47.75 | 80.20 | 21.95 | 43.88 | 62.06 | 17.35 |
| CLIP4Cir[†] | 12.82 | 36.83 | 48.19 | 81.91 | 24.12 | 46.47 | 63.07 | 18.47 |
| CompoDiff (ViT-L)[†] | 18.24 | 53.14 | 70.82 | 90.25 | 57.42 | 77.10 | 87.90 | 37.83 |
| CompoDiff [‡] (ViT-L)[†] | 19.37 | 53.81 | 72.02 | 90.85 | 59.13 | 78.81 | 89.33 | 39.25 |
| CompoDiff (ViT-G)[†] | **26.71** | **55.14** | **74.52** | **92.01** | **64.54** | **82.39** | **91.81** | **45.63** |
| Supervised | | | | | | | | |
| TIRG | 14.61 | 48.37 | 64.08 | 90.03 | 22.67 | 44.97 | 65.14 | 18.64 |
| TIRG + LastConv | 11.04 | 35.68 | 51.27 | 83.29 | 23.82 | 45.65 | 64.55 | 17.43 |
| MAAF | 10.31 | 33.03 | 48.30 | 80.06 | 21.05 | 41.81 | 61.60 | 15.68 |
| MAAF + BERT | 10.12 | 33.10 | 48.01 | 80.57 | 22.04 | 42.41 | 62.14 | 16.08 |
| MAAF-IT | 9.90 | 32.86 | 48.83 | 80.27 | 21.17 | 42.04 | 60.91 | 15.54 |
| MAAF-RP | 10.22 | 33.32 | 48.68 | 81.84 | 21.41 | 42.17 | 61.60 | 15.82 |
| CIRPLANT | 15.18 | 43.36 | 60.48 | 87.64 | 33.81 | 56.99 | 75.40 | 24.50 |
| CIRPLANT w/ OSCAR | 19.55 | 52.55 | 68.39 | 92.38 | 39.20 | 63.03 | 79.49 | 29.38 |
| ARTEMIS | 16.96 | 46.10 | 61.31 | 87.73 | 39.99 | 62.20 | 75.67 | 28.48 |
| Combiner | 38.53 | 69.98 | 81.86 | 95.93 | 68.19 | 85.64 | 94.17 | 53.36 |
| Pre-training with SynthTriplets18M & Fine-tuning with FashionIQ | | | | | | | | |
| ARTEMIS[†] | 18.85 | 51.44 | 68.01 | 91.93 | 38.85 | 62.00 | 77.68 | 28.85 |
| Combiner[†] | **39.99** | **73.63** | **86.77** | **96.55** | **68.41** | **86.12** | **94.80** | **54.20** |
| CompoDiff (ViT-L)[†] | 21.30 | 55.01 | 72.62 | 91.49 | 58.82 | 77.60 | 88.37 | 40.06 |
| CompoDiff (ViT-L + T5-XL)[†] | 22.35 | 54.36 | 73.41 | 91.77 | 62.55 | 81.44 | 90.21 | 42.45 |
| CompoDiff (ViT-G)[†] | 32.39 | 57.61 | 77.25 | 94.61 | 67.88 | 85.29 | 94.07 | 50.14 |

Table B.2: **Compasions on CIRR Test set.** Details are the same as Table B.1.

| Method | mAP@5 | mAP@10 | mAP@25 | mAP@50 |
|---|---|---|---|---|
| Previous zero-shot methods (without SynthTriplets18M) | | | | |
| CLIP + InstructPix2Pix | 1.83 | 2.10 | 2.37 | 2.44 |
| Pic2Word | 8.72 | 9.51 | 10.65 | 11.29 |
| SEARLE-XL | 11.68 | 12.73 | 14.33 | 15.12 |
| Zero-shot results with the models trained with SynthTriplets18M | | | | |
| ARTEMIS | 9.35 | 11.41 | 13.01 | 14.13 |
| Combiner | 9.77 | 12.08 | 13.58 | 14.11 |
| CompoDiff (ViT-L) | 12.55 | 13.36 | 15.83 | 16.43 |
| CompoDiff (ViT-L + T5-XL) | 12.31 | 13.51 | 15.67 | 16.15 |
| CompoDiff (ViT-G) | **15.33** | **17.71** | **19.45** | **21.01** |

Table B.3: **Compasions on CIRCO Test set.** Details are the same as Table B.1.

# C   More qualitative examples

**Open world zero-shot CIR comparisons with Pic2Word.**   We illustrate further comparisons with Pic2Word in Fig. C.1. Here, we can draw the same conclusions as in the main text: Pic2Word often cannot understand images or instructions (*e.g.*, ignores the "crowdedness" of the images, or retrieves irrelevant

| Method | Focus Attribute | | | Change Attribute | | | Focus Object | | | Change Object | | | Avg |
| | R@1 | R@2 | R@3 | R@1 | R@2 | R@3 | R@1 | R@2 | R@3 | R@1 | R@2 | R@3 | R@1 |
| --- | --- | --- | --- | --- | --- | --- | --- | --- | --- | --- | --- | --- | --- |
| Previous zero-shot methods (without SynthTriplets18M) | | | | | | | | | | | | | |
| Pic2Word | 15.65 | 28.16 | 38.65 | 13.87 | 24.67 | 33.05 | 8.42 | 18.01 | 25.77 | 6.68 | 15.05 | 24.03 | 11.16 |
| SEARLE-XL | **17.00** | **29.65** | **40.70** | 16.38 | 25.28 | 34.14 | 7.96 | 16.94 | 25.61 | 7.91 | 16.79 | 24.80 | 12.31 |
| Zero-shot results with the models trained with SynthTriplets18M | | | | | | | | | | | | | |
| ARTEMIS | 11.76 | 21.97 | 25.44 | 15.41 | 25.14 | 33.10 | 8.08 | 16.77 | 24.70 | 18.84 | 30.53 | 39.98 | 13.52 |
| Combiner | 14.11 | 24.08 | 34.12 | 18.39 | 28.22 | 37.13 | 8.49 | 16.70 | 25.21 | 18.72 | 30.92 | 40.11 | 14.93 |
| CompoDiff (ViT-L) | 13.50 | 24.32 | 36.11 | 19.20 | 28.64 | 37.20 | 8.11 | 16.39 | 25.08 | 18.71 | 31.69 | 40.55 | 14.88 |
| CompoDiff [‡] (ViT-L) | 13.01 | 25.01 | 36.75 | **19.88** | 28.64 | 37.18 | 8.60 | 16.28 | **25.85** | **18.94** | **31.80** | **40.58** | 15.11 |
| CompoDiff (ViT-G) | 14.32 | 26.70 | 38.41 | 19.72 | **28.78** | **37.39** | **9.18** | **19.11** | 25.77 | 18.71 | 31.71 | 40.22 | **15.48** |

Table B.4: **Compasions on GeneCIS Test set.** Details are the same as Table B.1.

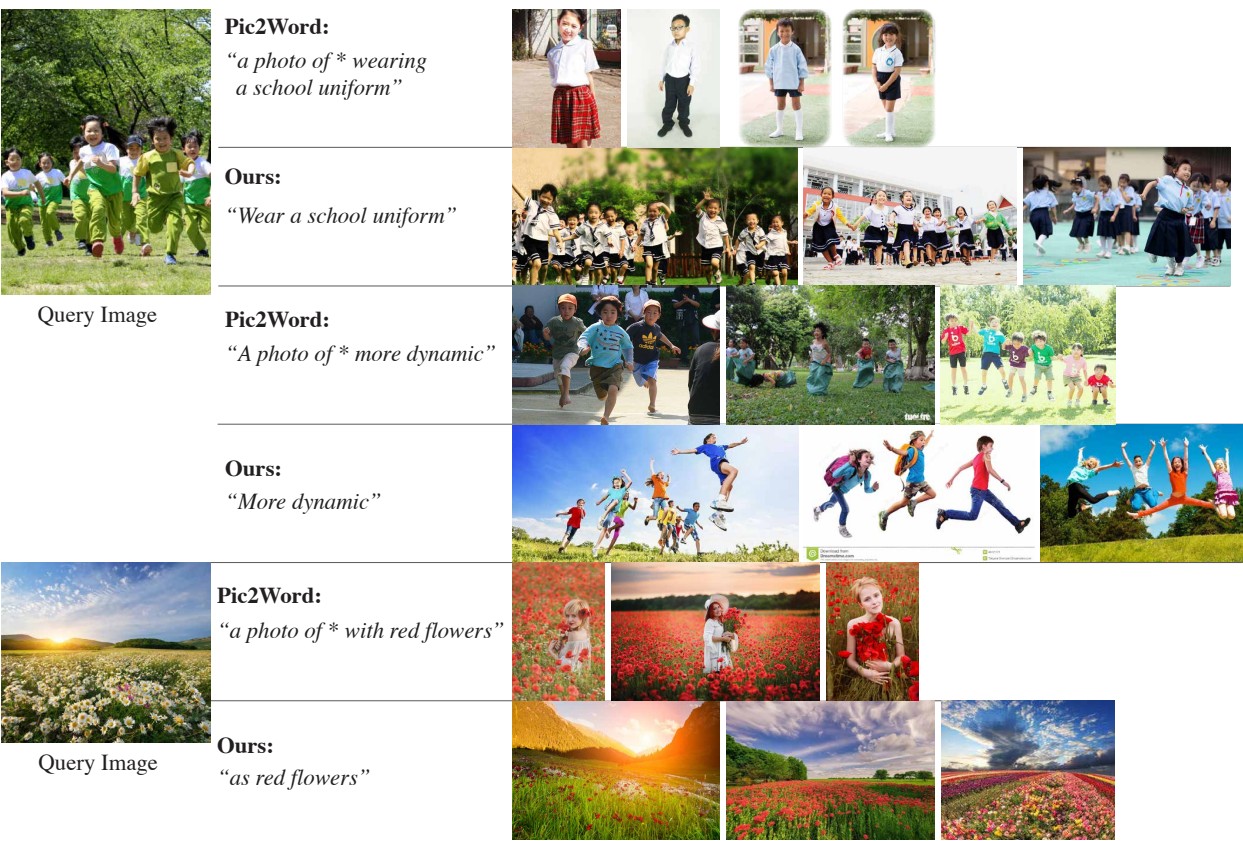

Figure C.1: **More qualitative comparison of zero-shot CIR for Pic2Word and CompoDiff.**

images such as images with a woman in the last example). All retrieved results in our paper were obtained using Pic2Word trained on the LAION 2B dataset.

**More versatile CIR examples on LAION.** We illustrate more qualitative examples of the composed features by CompoDiff, such as generated images by unCLIP retrieval results, in Fig. C.2, Fig. C.3, Fig. C.4, and Fig. C.5.

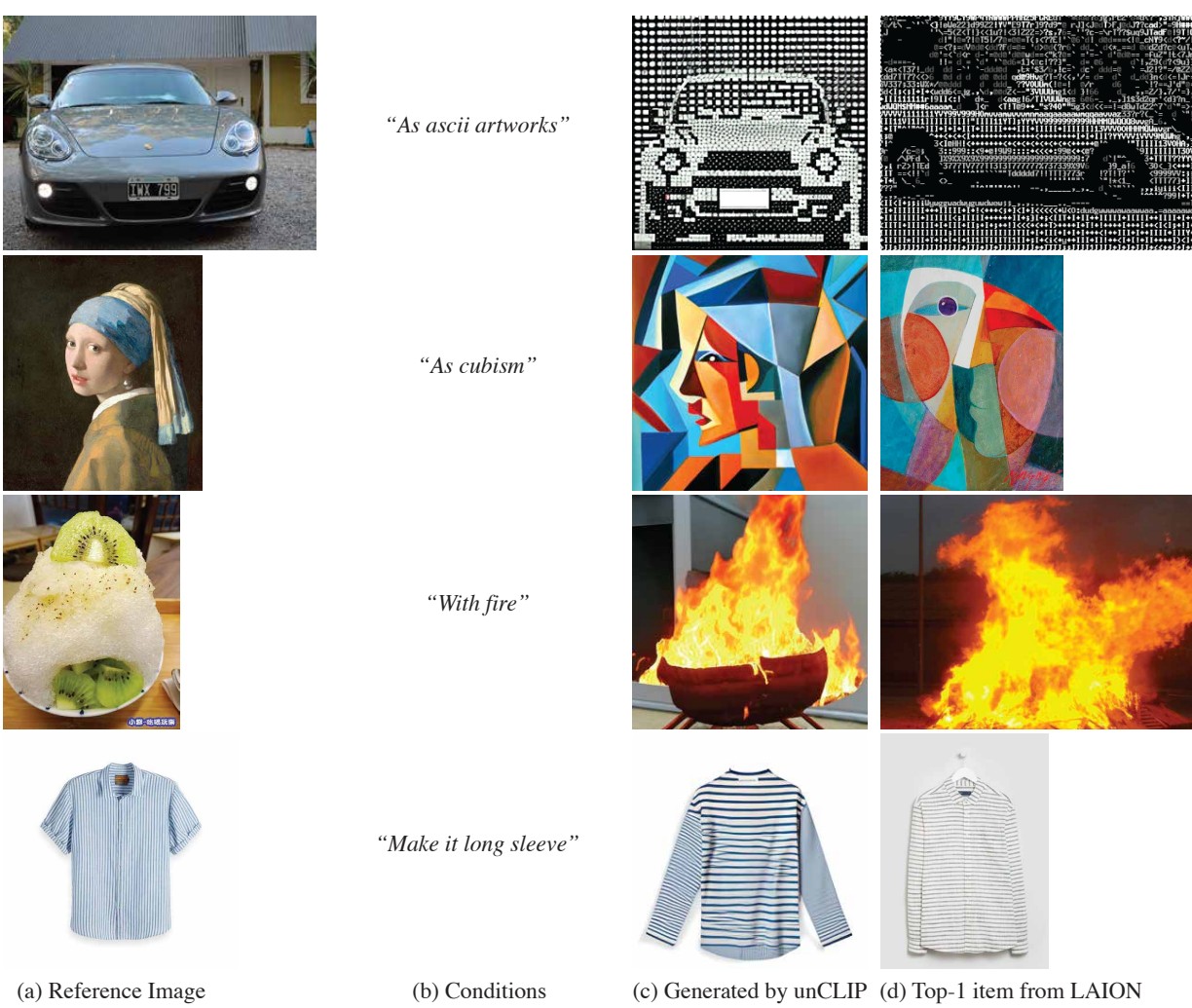

|  |  |  |  |
|---|---|---|---|
| (a) Reference Image | (b) Conditions | (c) Generated by unCLIP | (d) Top-1 item from LAION |

Figure C.2: **Generated vs. retrieved images by CompoDiff.** Using the transformed image feature by CompoDiff, Generated images using unCLIP and top-1 retrieved image from LAION.

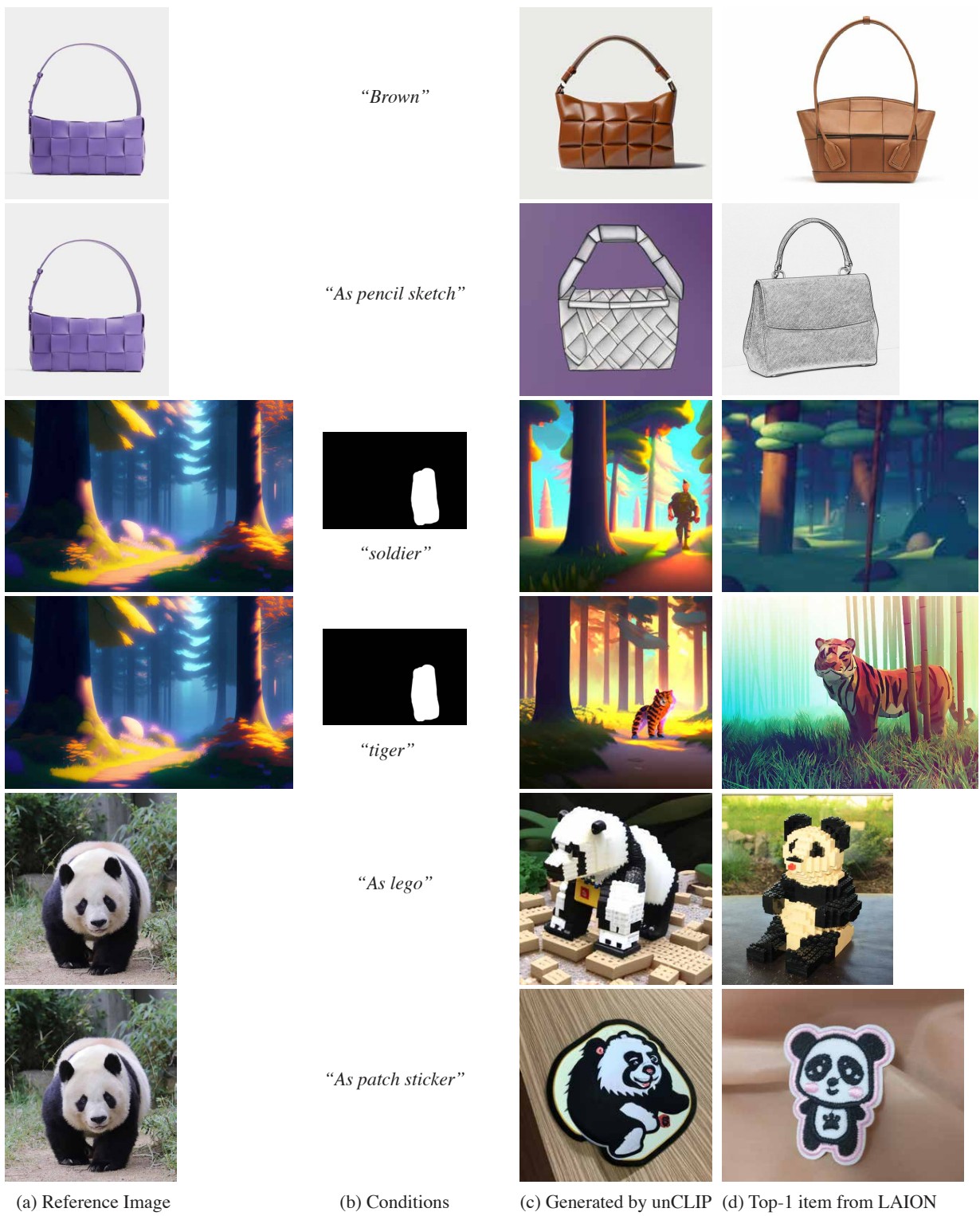

(a) Reference Image      (b) Conditions      (c) Generated by unCLIP    (d) Top-1 item from LAION

Figure C.3: **Generated vs. retrieved images by CompoDiff (Continue).**

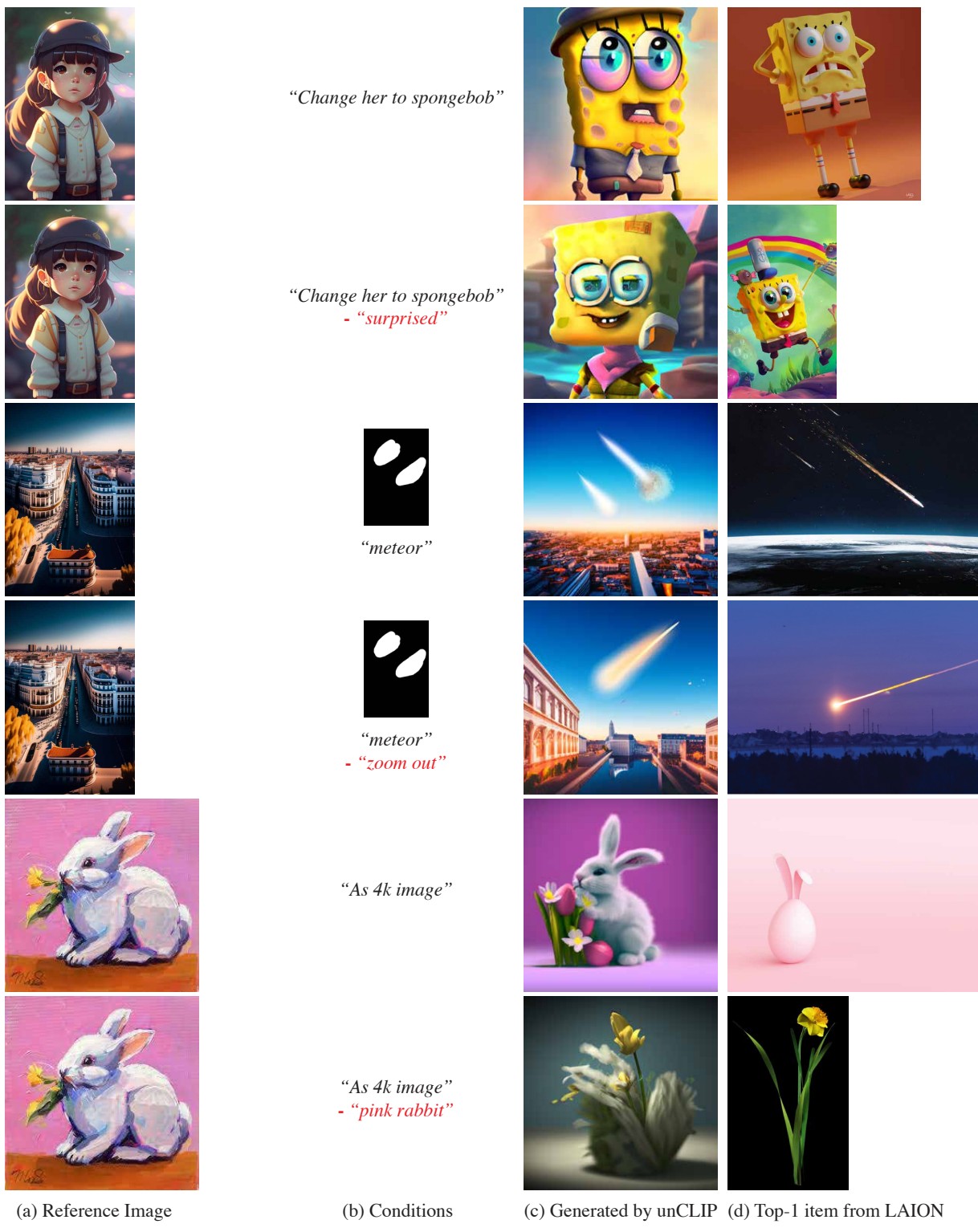

Figure C.4: **Generated vs. retrieved images by CompoDiff (Continue).**

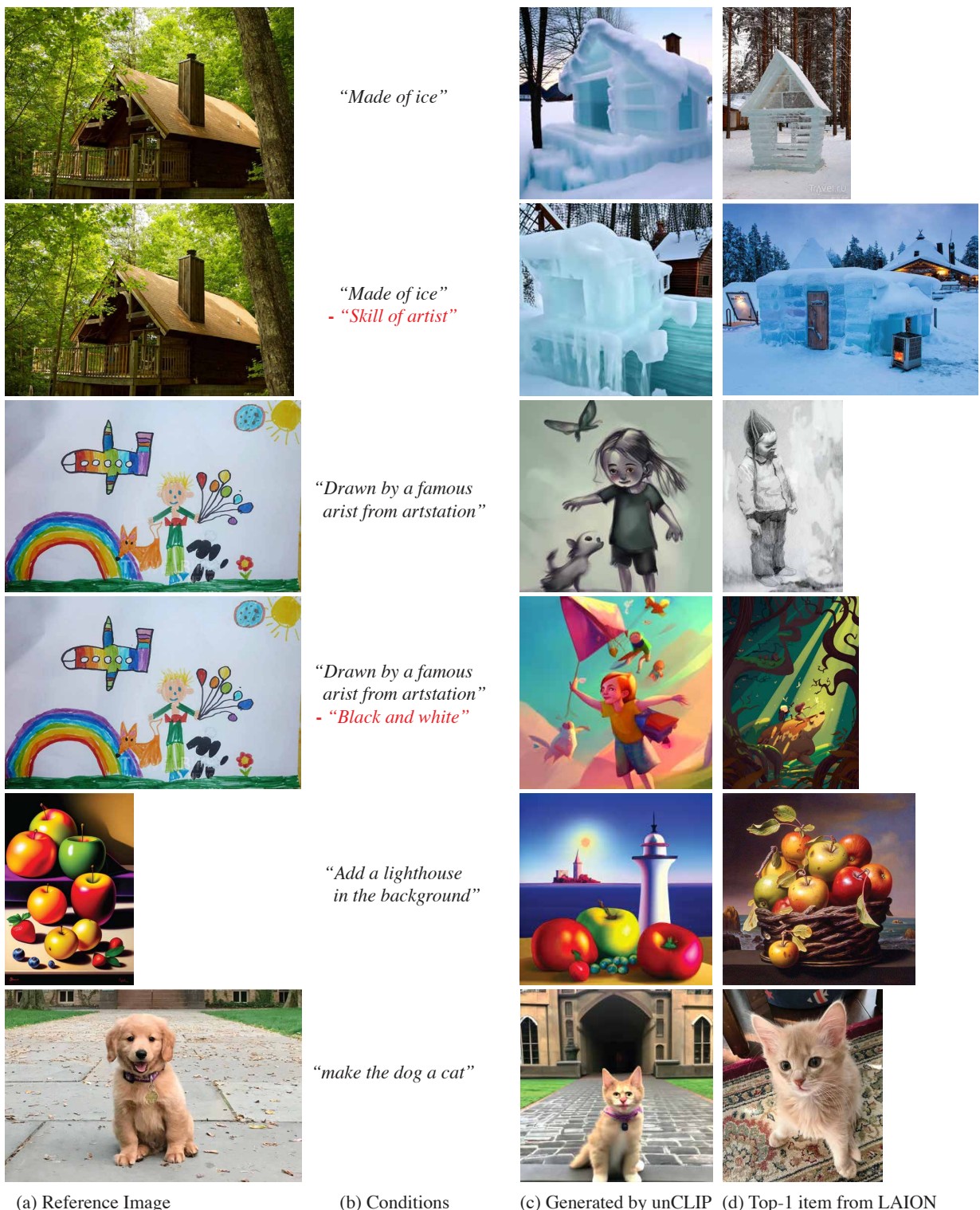

(a) Reference Image      (b) Conditions      (c) Generated by unCLIP    (d) Top-1 item from LAION

Figure C.5: **Generated vs. retrieved images by CompoDiff (Continue).**

