# OpenReview forum: "CompoDiff: Versatile Composed Image Retrieval With Latent Diffusion"
_TMLR — Accepted by TMLR_

### Review · Reviewer_SMao · 2024-04-14

**Summary Of Contributions:**

This paper aims to address zero-shot Composed Image Retrieval (ZS-CIR) task with latent representation via latent diffusion model (LDM). Prior methods usually adopt the encoder technique to encode the query image and other conditions (e.g., text) to a latent embedding for similarity retrieval, which may be limited to training conditions and not flexible during inference. Instead, this paper borrows ideas from the LDM-based image generation framework and classifier-free guidance technique, which enable flexible conditions and better performance. Besides, a large-scale synthetic dataset has been constructed based on various pretrained models and techniques (e.g., Prompt2Prompt). Abundant experiments have been done to show the effectiveness and superiority of the proposed method.

**Audience:**

Yes

**Broader Impact Concerns:**

The societal impact has been claimed in the paper. The potentially malicious samples have been filtered out during the creation of the proposed SynthTriplets18M dataset.

**Claims And Evidence:**

Yes

**Requested Changes:**

- There have been several techniques (e.g., distillation, samplers, consistency model) proposed to speed up the diffusion process. I recommend the authors to further alleviate inference costs to make the method more practical.
- GPU memory cost analysis is necessary to make this work more complete.

**Strengths And Weaknesses:**

Strengths
- This work enables flexible zero-shot Composed Image Retrieval during inference, thanks to the LDM framework. The input conditions could include the reference image, positive prompt, and negative prompt. The users can also re-weight different conditions to emphasize their true intentions.
- To better fit the CIR task, the denoiser is designed as a transformer-based model, and achieves good results in 5-10 denoising steps. The condition is integrated via a cross-attention mechanism to improve the inference speed.
- The constructed large-scale synthetic dataset and the corresponding automatic data generation techniques (e.g., triplet generation) could benefit the Composed Image Retrieval field or other related tasks. The experiment shows that Combiner trained with SynthTriplets18M may be boosted in the zero-shot setting.
- The dataset statistics analysis in Fig. 7 & Fig. 8 are good for showing the domain gap between different datasets. The dataset scale experiment in Table 6 validates the scalability of the proposed method to some extent.

Weaknesses
- The inference cost is the biggest limitation. Although it is claimed that the test speed can be improved via a trade-off between inference speed and performance, the current speed is still 10x slower than the encoder-only methods. For example, in Table 4, Combiner can achieve comparable performance with x20 test speed. For the Image Retrieval task, test speed is very important for practical usage.
- The GPU memory cost analysis is also important for practical usage, which is missing in the main paper.

---

> ### Author Response · Authors · 2024-05-18
>
> We appreciate your constructive comments. We addressed all the concerns raised in the comment and updated our paper accordingly. Please let us know if your concerns remain after reading the revised paper.
>
> ## Slow inference speed and advanced techniques for boosting up diffusion models
>
> Thanks for your constructive comment. First, we would like to emphasize that the current latency, 100ms, is slightly above the bar for practical retrieval systems.
>
> We also absolutely believe that boosting up our CompoDiff diffusion model will bring a huge benefit. We tried to apply distillation approaches as the reviewers commented, but we believe that huge and non-trivial efforts are needed, which can be considered a new contribution. More specifically, our model is based on both the text-to-image diffusion model (stage 1) and the image-to-image but text-conditioned diffusion model (stage 2). On the other hand, most of the existing boosting-up techniques, as far as our knowledge, are applicable to text-free diffusion models (i.e., the original Gaussian diffusion models without conditions). We found a paper that can apply a distillation for guided DMs [A], but we found that they do not provide any official implementation, and it is very difficult to reproduce their results on top of our framework.
>
> [A] On Distillation of Guided Diffusion Models, Chenlin Meng, Robin Rombach, Ruiqi Gao, Diederik P. Kingma, Stefano Ermon, Jonathan Ho, Tim Salimans,  CVPR 2023
>
> Similarly, the consistency model needs a different training procedure, and again, it is nontrivial to apply the consistency model approach to our framework. We also would apply more recent faster latent diffusion models, such as SD-XL turbo, for the backbone of our method. These approaches will need additional training and corresponding contributions, thus, we leave this for future work.
>
> Instead, (1) we report sampling steps 2-4 in Tab 4 – step 4 is faster without a significant performance drop, and (2) we add a related discussion for recent advances of diffusion models in terms of sampling speed in Section 5.6. We hope our new discussion can mitigate the reviewer’s concern.
>
> ## GPU memory cost analysis
>
> Thanks for your feedback. We added information in Table 4. We also clarify that ARTEMIS resource information is not comparable in inference scenario because its forward path needs a triplet of reference image, instruction, and target image, not an image-instruction pair.

---

> > ### Comment · Reviewer_SMao · 2024-05-25
> > **Thanks for the rebuttal**
> >
> > I have read the rebuttal and other reviews. My concerns about the inference speed and GPU memory costs have been addressed. The proposed method leverages the image generative model for the controllable image retrieval tasks and achieves better performance than prior works. The inference speed and GPU memory are acceptable considering the added controllability.
> >
> > This work may also inspire other related CV fields to utilize the priors in large-scale pretrained generative models. I would vote for acceptance.

---

> > > ### Author Response · Authors · 2024-05-26
> > >
> > > We truly appreciate your response!
> > >
> > > Authors

---

### Review · Reviewer_5Q5f · 2024-05-03

**Summary Of Contributions:**

This paper proposes CompoDiff, a diffusion-based model for solving zero-shot Composed Image Retrieval (ZS-CIR).
- CompoDiff can handle various conditions, such as negative text and image mask conditions.
- CompoDiff can control the strength of conditions and the trade-off between inference speed and performance.

This paper also proposes SynthTriplets18M, which are synthesized triplets for training.

The experiments support the effectiveness of CompoDIff.
- Achieved significant performance on FashionIQ, CIR, CIRCO, and GeneCIS datasets.
- Qualitative performance on ZS-CIR on a billion-scale database, or image decoding using the unCLIP generator.

**Audience:**

Yes

**Claims And Evidence:**

Yes

**Requested Changes:**

- Justify the comparison with ARTEMIS and Combiner which do not use LIAON-2B datasets.

- In Fig.3, bottom of Stage 1, there are two text conditions.　In my understanding, the change condition, such as "Change the chickadee to a rooster," is not utilized in Stage 1. Could you explain how these two text conditions are employed in Stage 1?

- Add CIRCO and GenCIS datasets to Fig. 8.

- Table3 with Arch RN50, the performance of CompoDIff is lower than that of Combiner on GeneCIS.
Could you explain this reason?

- In Sec.5.7, Dall-e2 Prior model is referred as (Ramesh et al., 2022). However, Table 5 refers (Lee et al., 2022) which is the reference of footnote 4. To prevent confusion, add the citation ( Lee et al., 2022) in the main text.

**Strengths And Weaknesses:**

Strengths.
- The proposed CompoDiff can handle negative text conditions and mask conditions.
- The paper is well-written and addresses its relation to many state-of-the-art works.
- SynthTriplet18M dataset contains many triplets that are automatically generated.
- The experiment is extensive and promising.
    - CompoDiff outperforms state-of-the-art methods on this problem.
     - There are several ablation studies and detailed analysis
     - Abundant example images are shown.

Weakness.
- The proposed method is a combination of existing methods.
- The comparison with ARTEMIS and Combiner seems unfair because they did not utilize the LAION-2B datasets for training.

---

> ### Author Response · Authors · 2024-05-18
>
> Thank you for your thoughtful review. We address all the raised concerns in the comment and updated our paper accordingly. Please let us know if your concerns still remain after reading the revised paper.
>
> ## Fair comparison with ARTEMIS and Combiner
>
> We first emphasize that ARTEMIS and Combiner are impossible to use LAION-2B dataset because they are only trainable with triplets (reference image, edit instruction and target image), while LAION is a pairwise dataset (image and its corresponding caption).
>
> We also would like to emphasize that our stage 1 can be viewed as pre-training a latent diffusion model using LAION-2B. In this point of view, Combiner also has a similar stage for pre-training the CLIP backbone with paired dataset. If we take this point of view, we can argue that Combiner is also trained on a pairwise dataset, which is now comparable with CompoDiff.
>
> In summary, ARTEMIS and Combiner are not possible to be trained on paired datasets, where LAION is a paired dataset. From another point of view, because Combiner needs a CLIP backbone trained on a large-scale paired dataset, we also can argue that Combiner already uses a large-scale paired dataset, which is comparable with LAION-2B used by CompoDiff.
>
> We clarified our argument in Section 5.3, highlighted by magenta.
>
> ## Combination of existing methods
>
> Although the main components of our method and dataset construction are from the existing methods, our method is the first method to apply image editing methods to composed image retrieval tasks. Furthermore, our diffusion model has its novelty in (1) architecture and (2) new conditions. As the architecture contribution, we first introduce an efficient transformer-based diffusion architecture rather than the popular U-Net structure. It is clarified in Section 3.1 and Section 5.7 Analysis “Denoising Transformer ablation”. We also introduce new mask conditions that have not been used before for training latent diffusion models. We extract masks from images using a zero-shot mask predictor and use the mask condition to train our model.
>
> Furthermore, our dataset synthesis process also has a non-trivial novelty compared to InstructPix2Pix. First, we propose a simple yet effective template-based text triplet construction method. Compared to LLM-generated triplets used by InstructPix2Pix, our template-based construction can contain more diverse and realistic captions from actual captions from the LAION dataset. Also, our template-based construction can make diverse caption triplets that could rarely happen (e.g., the pak choi example in Figure 6).
>
> In summary, we believe that our approach is sufficiently novel in (1) applying editing models and diffusion models to retrieval tasks, (2) new diffusion architecture and conditions, and (3) a simple yet effective template-based text triplet construction method for the dataset construction.
>
> #### Performance comparison with Combiner RN50
>
> Thanks for the question. We first emphasize that CompoDiff outperforms Combiner in all other datasets, and performs better in average. When the number of the zero-shot tasks becomes larger, it is not guaranteed that the best performed model can achieve the best performance across all the tasks [A].
>
> [A] Inherent Trade-Offs between Diversity and Stability in Multi-Task Benchmark, G Zhang, M Hardt, ICML 2024
>
> Second, it could be the problem of the dataset. In our GeneCIS evaluation, we use very naive instruction texts, as same as the task itself, namely, "focus attribute", "change attribute", "focus object", and "change object". This means that all retrievals performed for each sub-task have the same text instruction, and the instructions do not contain explicit information about the target image. As shown in the updated Fig 8, the token distribution of GeneCIS instructions is very different from the other datasets. We presume that the visual feature transform by CompoDiff could underperform under such short, noisy, and less informative instructions. We believe that CompoDiff would perform better than Combiner in GeneCIS if we use more realistic instruction queries containing more information about the target image.
>
> ## Others
>
> - It is our mistake. Stage 1 only has a single text condition, the stage 1 in figure 3 is not correct. We corrected the figure in the revision. Thanks for your comment!
> - We updated Fig 8 to contain CIRCO and GeneCIS. Thanks for your constructive feedback.
> - The correct reference for Dall-e2 Prior model is Ramesh et al., 2022. Lee et al., 2022 is for the Karlo model, which is a community version of Dall-e2 model. We clarified it in Table 5 of the revised version.

---

### Review · Reviewer_X9BB · 2024-05-12

**Summary Of Contributions:**

This paper investigates an interesting task in image retrieval called Composed Image Retrieval (CIR), where the customer needs a search engine that can process composed queries. This paper treats the CIR task as a conditional image editing task on the latent space. It proposes a diffusion-based model with latent diffusion, called CompoDiff, for solving zero-shot Composed Image Retrieval (ZS-CIR). It also creates a large and diverse dataset named SynthTriplets18M, consisting of 18.8M triplets of images, modification texts, and modified images. The experiments are well designed to show the effectiveness of the proposed method and datasets.

**Audience:**

Yes

**Broader Impact Concerns:**

I believe this paper addressed the Broader Impacts.

**Claims And Evidence:**

Yes

**Requested Changes:**

1.the description of  $w_I$ and $w_T$ following Eqn.4
2.Amplificaiton the importances of the Composed Image Retrieval (CIR), and shows the unique difficulties of CIR, compare to image edit/generation.

**Strengths And Weaknesses:**

**Strengths:**

The motivation and clarification are clear, and I can understand the details following the descriptions. The proposed CompoDiff, a diffusion-based model with latent diffusion, is effective for solving zero-shot Composed Image Retrieval (ZS-CIR). The constructed dataset, named SynthTriplets18M, is valuable for the community in image retrieval. The experiments are sufficient to show the effectiveness of the proposed method and datasets.


**Weaknesses:**

1.I am not sure whether the contribution is significant, due to the image edit tasks has achieved great success. Based on my understanding, the task of image retrieval is easier than the task of image generation. When there are so many great works in image generation (e.g., controllable edit and generation), I am not sure whether the progress this paper contribute in ZS-CIR is significant, especially it is based on the techniques of image generation tasks.

2.This paper points out that the drawbacks of the existing CIR benchmarks are not scalable for “real-world queries”. However, I find the images of the constructed SynthTriplets18M datasets are also not “real-world images”, which are generated by generative models. This leaves the debate that whether the SynthTriplets18M can represent “real-world queries”.

Other minors:
In Eqn.4, the weight $w_I$ and $w_T$ should be illustrated in the following description, rather than putting in another paragraph. When I take took the first look, I thought $w_I$/$w_t$ was a function.

---

> ### Author Response · Authors · 2024-05-18
>
> We appreciate your professional review. We addressed all the raised concerns in the comment and updated our paper accordingly. Please let us know if your concerns still remain after reading the revised paper.
>
> ## Contribution compared to image editing
>
> Image retrieval is not always easier than image generation (more precisely, instruction-based image editing). Note that the purpose of image editing is to find any image that satisfies the changes due to the instructions. For example, assume there exists a picture of a dog on grass, and our desired instruction is "change the dog to a cat". Even if we assume the perfect image generator satisfies the conditioned editing, it may not be the best method for retrieval. This is because our desired behavior for a good CIR method is to search for all possible images that satisfy the given instructions. However, if we just edit an image and use the image as the query image, the retrieval results will not fully reflect the instruction. In other words, the purpose of image editing is “precision”, while the purpose of image retrieval is “recall”; it makes the difference between image editing and retrieval.
>
> Table 3 supports this claim. We apply InstructPix2Pix to edit images with the given instructions. Then, we extract features using the pre-trained CLIP feature extractor (which is the same feature extractor as the other comparison methods). As shown in the table, naively using an image editing method cannot help composed image retrieval task.
>
> Therefore, an instruction-conditioned image editing task is not superior to a composed image retrieval task, and we need a retrieval-oriented model design principle. In our paper, this design principle is (1) using a latent diffusion rather than image-level diffusion and (2) introducing retrieval-specific conditions, such as mask conditions.
>
> We added the related discussion in the 3rd paragraph of the introduction, highlighted by magenta.
>
> ## SynthTriplets18M samples cannot represent real-world samples
>
> As the reviewer's comment, although we use a state-of-the-art generation method for synthesizing our dataset, it will not be the same as the real-world images.
>
> However, our argument is not about "realistic images". We argue that SynthTriplets18M instructions are more realistic and generalizable than those of FashionIQ and CIRR. For example, FashionIQ only contains text instructions for fashion items (e.g., "Is blue and has stripes") and CIRR has very noisy and somewhat not helpful text instructions (e.g., "Same environment different species" "The target photo is of a lighter brown dog walking in white gravel along a wire and wooden fence"). On the other hand, our generated instructions are more realistic queries as shown in Figure 9 (e.g., "make it spring", "4k high resolution images is the new choice"). Therefore, we argue that a model trained on our CIR triplets can learn meaningful and generalizable representations, showing great ZS-CIR performances.
>
> We add the related discussion in Section 4, highlighted by magenta.
>
> ##  $w_I$ and $w_T$
>
> Thanks for the comment! We updated the paragraph as the suggestion (highlighted by magenta).

---

### Decision · Action_Editor_BFSi · 2024-06-26

**Recommendation:** Accept as is

**Comment:**

All reviewers have acknowledged the contribution of this paper, as well as the revision made in response to the reviews. All reviewers recommended to accept the paper. This paper proposes CompoDiff, a latent diffusion based method for solving composed image retrieval in the zero-shot scenario. It also introduces SynthTriplets18M, a new synthetic dataset with 18.8 million reference images, conditions, and
corresponding target image triplets. These two contributions are good addition to the literature, expanding the scope of generative models to retrieval tasks.

**Audience:**

TMLR's audience will be generally interested in the findings of this paper, especially the verification that large diffusion models can enable composed image retrieval in the zero-shot scenario.

**Claims And Evidence:**

Based on the reviews, all claims made in the submission are well supported by model design and empirical evidence.